# Control of tissue development and cell diversity by cell cycle-dependent transcriptional filtering

**Maria Abou Chakra, Ruth Isserlin, Thinh N Tran, Gary D Bader***

The Donnelly Centre, University of Toronto, Toronto, Canada

**Abstract** Cell cycle duration changes dramatically during development, starting out fast to generate cells quickly and slowing down over time as the organism matures. The cell cycle can also act as a transcriptional filter to control the expression of long gene transcripts, which are partially transcribed in short cycles. Using mathematical simulations of cell proliferation, we identify an emergent property that this filter can act as a tuning knob to control gene transcript expression, cell diversity, and the number and proportion of different cell types in a tissue. Our predictions are supported by comparison to single-cell RNA-seq data captured over embryonic development. Additionally, evolutionary genome analysis shows that fast-developing organisms have a narrow genomic distribution of gene lengths while slower developers have an expanded number of long genes. Our results support the idea that cell cycle dynamics may be important across multicellular animals for controlling gene transcript expression and cell fate.

## Introduction

A fundamental question in biology is how a single eukaryotic cell (e.g., zygote, stem cell) produces the complexity required to develop into an organism. A single cell will divide and generate many progeny, diversifying in a controlled and timely manner (*Mueller et al., 2015*) to generate cells with very different functions than the parent, all with the same genome (*Wilmut et al., 1997*). Many regulatory mechanisms coordinate this process, but much remains to be discovered about how it works (*Zoller et al., 2018*). Here, we explore how cell cycle regulation can control gene transcript expression timing and cell fate during tissue development.

The canonical view of the cell cycle is a timely stepwise process. Typically, the complete cell cycle is divided into four phases: first gap phase (G1), synthesis phase (S), second gap phase (G2), and mitotic phase (M). The length of each phase determines how much time a cell allocates for processes associated with growth and division. However, the amount of time that is spent in each phase frequently differs from one cell type to another within the same organism. For example, some cells experience fast cell cycles, especially in early embryogenesis. Organisms such as the fruit fly (*Drosophila melanogaster*) and the worm (*Caenorhabditis elegans*) exhibit cell cycle durations as short as 8–10 min (*Edgar et al., 1994*; *Foe, 1989*). Cell cycle duration also changes over development (*Figure 1* and *Supplementary file 1*). For example, it increases in mouse (*Mus musculus*) brain development from an average of 8 hr at embryonic day 11 (E11) to an average of 18 hr by E17 (*Furutachi et al., 2015*; *Takahashi et al., 1995a*).

Interestingly, cell cycle duration can act as a transcriptional filter that constrains transcription (*Rothe et al., 1992*; *Shermoen and O'Farrell, 1991*). In particular, if the cell cycle progresses relatively fast, transcription of long genes will be interrupted. In typical cells, the gene transcription rate is between 1.4 and 3.6 kb per minute (*Ardehali and Lis, 2009*). Thus, an 8 min cell cycle would only allow transcription of the shortest genes, on the order of 10 kb measured by genomic length,

**\*For correspondence:**
gary.bader@utoronto.ca

**Competing interests:** The authors declare that no competing interests exist.

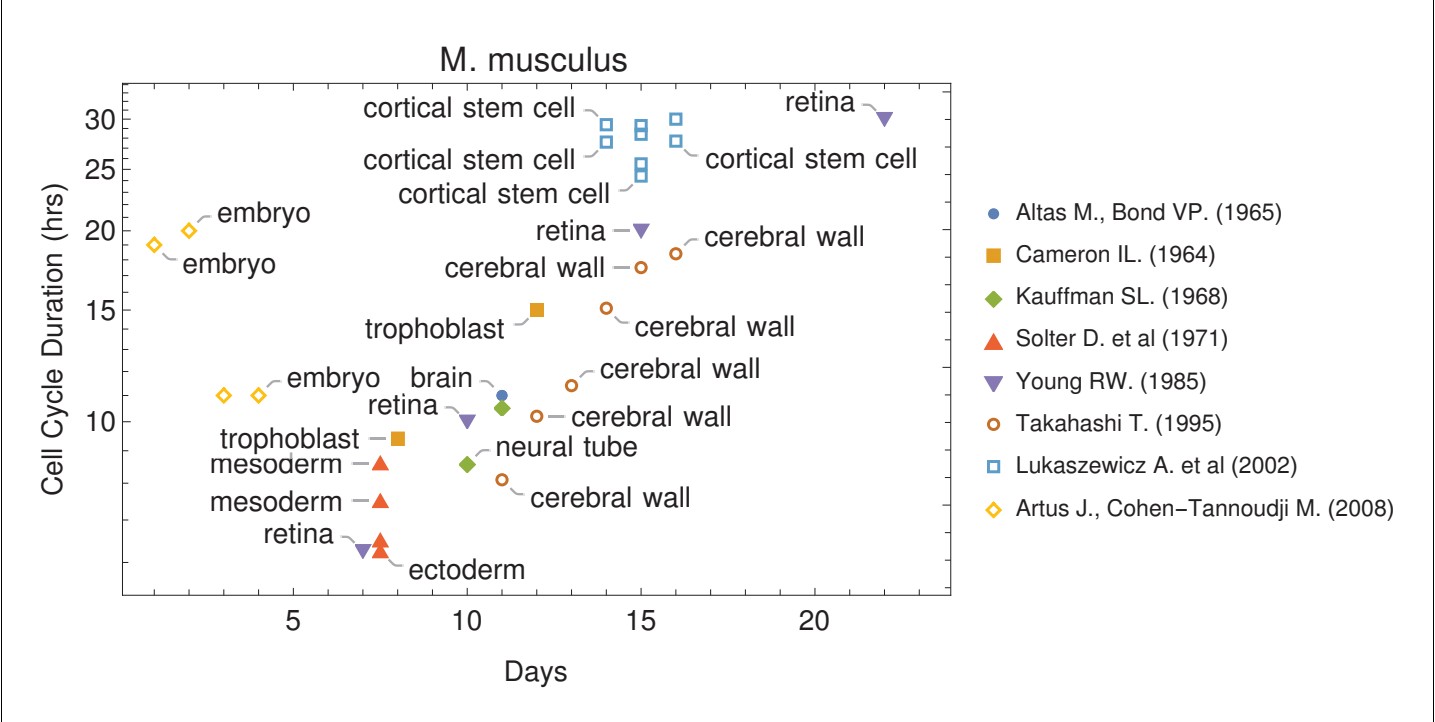

**Figure 1.** Cell cycle duration changes during mouse development. The data was curated from several publications (PubMed identifiers: 5859018, 14105210, 5760443, 5542640, 4041905, 7666188, 12151540, 18164540), shown in the legend as authors and (year). For other species and tissues, see *Supplementary file 1*.

including introns and exons, whereas a 10 hr cell cycle would allow transcription of genes as long as a megabase on the genome.

Cell cycle-dependent transcriptional filtering has been proposed to be important in cell fate control (*Bryant and Gardiner, 2016*; *Swinburne and Silver, 2008*). Most multicellular eukaryotic animals start embryogenesis with short cell cycle durations and a limited transcription state (*O'Farrell et al., 2004*) with typically short zygotic transcripts (*Heyn et al., 2014*). These cells allocate the majority of their cycle time to S-phase (synthesis), where transcription is inhibited (*Newport and Kirschner, 1982a*), and M-phase (division), with little to no time for transcription in the gap phases. However, as the cell cycle slows down, time available for transcription increases (*Edgar et al., 1986*; *Newport and Kirschner, 1982a*; *Newport and Kirschner, 1982b*), enabling longer genes to be transcribed (*Djabrayan et al., 2019*; *Shermoen and O'Farrell, 1991*; *Yuan et al., 2016*).

We asked what effects cell cycle-dependent transcriptional filtering may have over early multicellular organism development. Through extensive mathematical simulations of developmental cell lineages, we identify the novel and unexpected finding that a cell cycle-dependent transcriptional filter can directly influence the generation of cell diversity and can provide fine-grained control of cell numbers and cell-type ratios in a developing tissue. Our computational model operates at single-cell resolution, enabling comparison to single-cell RNA-seq (scRNA-seq) data captured over development, supporting our model by showing similar trends. Our model also predicts genomic gene length distribution and gene transcript expression patterns that are consistent with a range of independent data. Our work provides new insight into how cell cycle parameters may be important regulators of cell-type diversity over development.

## Results

### Computational model of multicellular development

We model multicellular development starting from a single totipotent cell that gives rise to many progeny, each with its own transcriptome (*Figure 2*). We developed a single-cell resolution agent-based computational model to simulate this process (see Materials and methods). Each cell behaves according to a set of rules, and cells are influenced solely by intrinsic factors (e.g., number of genes in the genome, gene length, transcript levels, and transcription rate). We intentionally start with a simple set of rules, adding more rules as needed to test specific mechanisms. Our analysis is limited

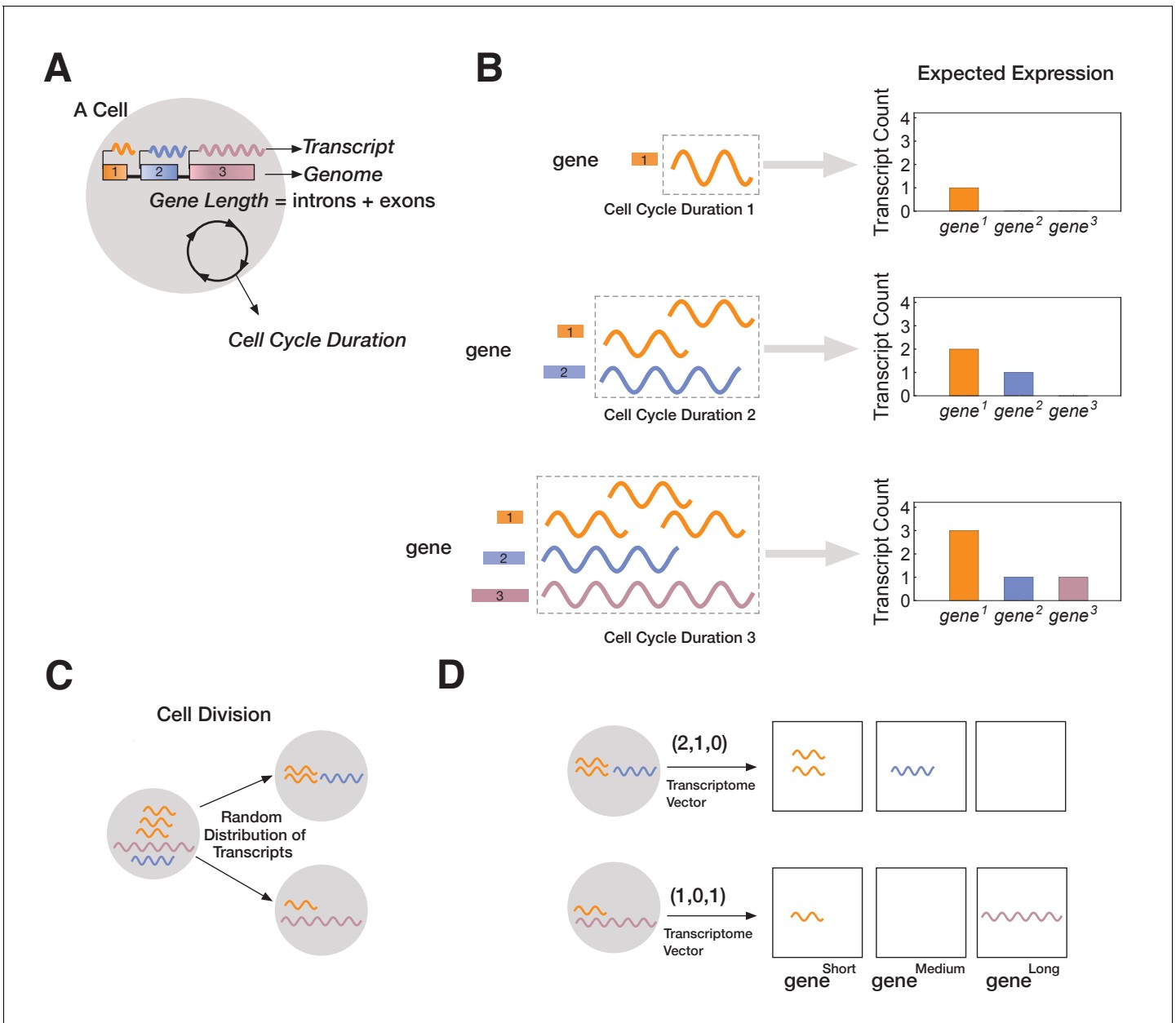

**Figure 2.** A novel mathematical model of cell lineage generation. (**A**) A single cell is defined by a given number of genes in its genome as well as their gene lengths (e.g., three genes, gene[1] < gene[2] < gene[3]). Cell cycle duration defines the time a cell has available to transcribe a gene. (**B**) For example, a cell with cell cycle duration = 1 hr will only enable transcription of gene[1]; cell cycle duration = 2 hr will enable transcription of gene[1] and gene[2]; cell cycle duration = 3 hr enables transcription of all three genes. (**C**) Our model assumes that transcripts passed from parental cell to its progeny will be randomly distributed during division (M-phase). (**D**) Each cell is characterized by its transcriptome, represented as a vector.

to pre-mRNA transcript expression, and we do not consider other gene expression-related factors, such as splicing, translation, or gene-gene interactions. We also omit external cues (e.g., intercellular signaling or environmental gradients) to focus on the effects of intrinsic factors.

In our model, each cell is characterized by a fixed genome containing a set of G genes $(\text{gene}_1, \text{gene}_2, \ldots \text{gene}_G)$ shown in (**Figure 2A**). Each $\text{gene}_i$ is defined by a length, $L_i$ (in kb), and in all our simulations each gene is assigned a different length ($L_1 < L_2 < \ldots L_G$). Since each $\text{gene}_i$ has a unique length, $L_i$, we label genes by their length ($\text{gene}_i^{L_i} = \text{gene}^{L_i}$; e.g., gene³ is a gene of length 3 kb). We assume transcription time for $\text{gene}_i$ is directly proportional to its length, $L_i$. In the model, each $\text{cell}_j$ is initialized with a cell cycle duration ($\Gamma_{\text{cell}}$), which represents the total time available for gene transcription (see Materials and methods). For example, we can initialize $\text{cell}_1$ with a three-gene genome $(\text{gene}^1, \text{gene}^2, \text{gene}^3)$, where $L = (1\text{kb}, 2\text{kb}, 3\text{kb})$ and a cell cycle duration $\Gamma_1$ of 1 hr. We fix transcription rate, $\lambda$, to 1 kb/hr for all genes (though this assumption can be relaxed without changing our results; **Figure 3** and **Figure 3—figure supplement 1**). As transcription progresses for all genes, $\text{cell}_1$ will only express $\text{gene}^1$. Increasing cell cycle duration, $\Gamma_{\text{cell}}$, will allocate more time for transcription, allowing longer genes to be transcribed. For example, if we initialize $\text{cell}_2$ with a cell cycle duration $\Gamma_2 = 3$ hr, $\text{cell}_2$ will express all three genes, with time to make three copies of $\text{gene}^1$ (**Figure 2B**). We assume that RNA polymerase II re-initiation occurs along the gene, a distance $\Omega$ apart (**Figure 3—figure supplement 1**).

Once transcription is complete, the cell enters M-phase, during which it divides, and expressed transcripts are randomly distributed to the two progeny cells (**Figure 2C**). This is the main stochastic component in our model. We assume that transcription begins anew at the start of the cell cycle (i. e., all transcripts from a gene that cannot be finished in one cycle are eliminated), modeling the known degradation of incomplete nascent transcripts in M-phase (**Shermoen and O'Farrell, 1991**). Relaxing our assumption to include parental transcript inheritance and decay (**Sharova et al., 2009**), where a proportion of inherited parental transcripts remain after each cell division, does not change our overall results (**Figure 3—figure supplement 2**). All individual cells and their transcriptomes are tracked over the course of the simulation, enabling single-cell resolution analysis. Transcriptomes are stored as vectors containing the total number of transcripts per gene. For instance, $\text{cell}_2$ may have a transcriptome of (3,1,1), indicating that three genes are expressed, with $\text{gene}^1$ expressed at three transcripts per cell and the other two genes expressed at one transcript per cell (**Figure 2D**).

## Model prediction: cell cycle duration influences transcript count – short genes generate more transcripts than longer genes

We begin by examining how a transcriptional filter impacts transcript counts, as controlled by cell cycle duration. Shorter cell cycles will interrupt long gene transcription, resulting in relatively high expression of short gene transcripts and low expression of long gene transcripts. Our computational simulations generate this expected pattern (**Figure 3A**). Each simulated cell transcriptome is divided into three bins containing short, medium, and long genes, and then each bin is summarized with an average transcript count. In simulations, bins with short genes exhibit the highest average transcript count levels. As cell cycle duration increases, more cells show an increase in transcript count of longer genes; the trend is consistent for various genome sizes and gene length distributions (**Figure 3A** and **Figure 3—figure supplement 3**).

scRNA-seq has recently been used to profile mRNA expression of thousands of cells for one cell type (microglia) across multiple species (**Geirsdottir et al., 2019**) or for multiple embryonic developmental time points in one species, such as *Xenopus tropicalis* (**Briggs et al., 2018**) and *Danio rerio* (**Kimmel et al., 1995**; **Wagner et al., 2018**), or tissue, such as mouse neural cortex (**Yuzwa et al., 2017**). We analyzed these data in the same manner as our model (**Figure 3B** and **Figure 3—figure supplement 4**) and found that, in general, short genes have a higher mRNA expression level than longer genes within a cell. Thus, gene mRNA expression patterns from a range of scRNA-seq data sets, including developmental time courses, are compatible with our model prediction.

## Model prediction: cell cycle duration can control cell diversity

We next asked how three major model parameters (cell cycle duration, maximum gene length, and number of genes in the genome) can influence the generation and control of cell diversity observed during normal multicellular development. We conducted simulations for a single-cell division step for

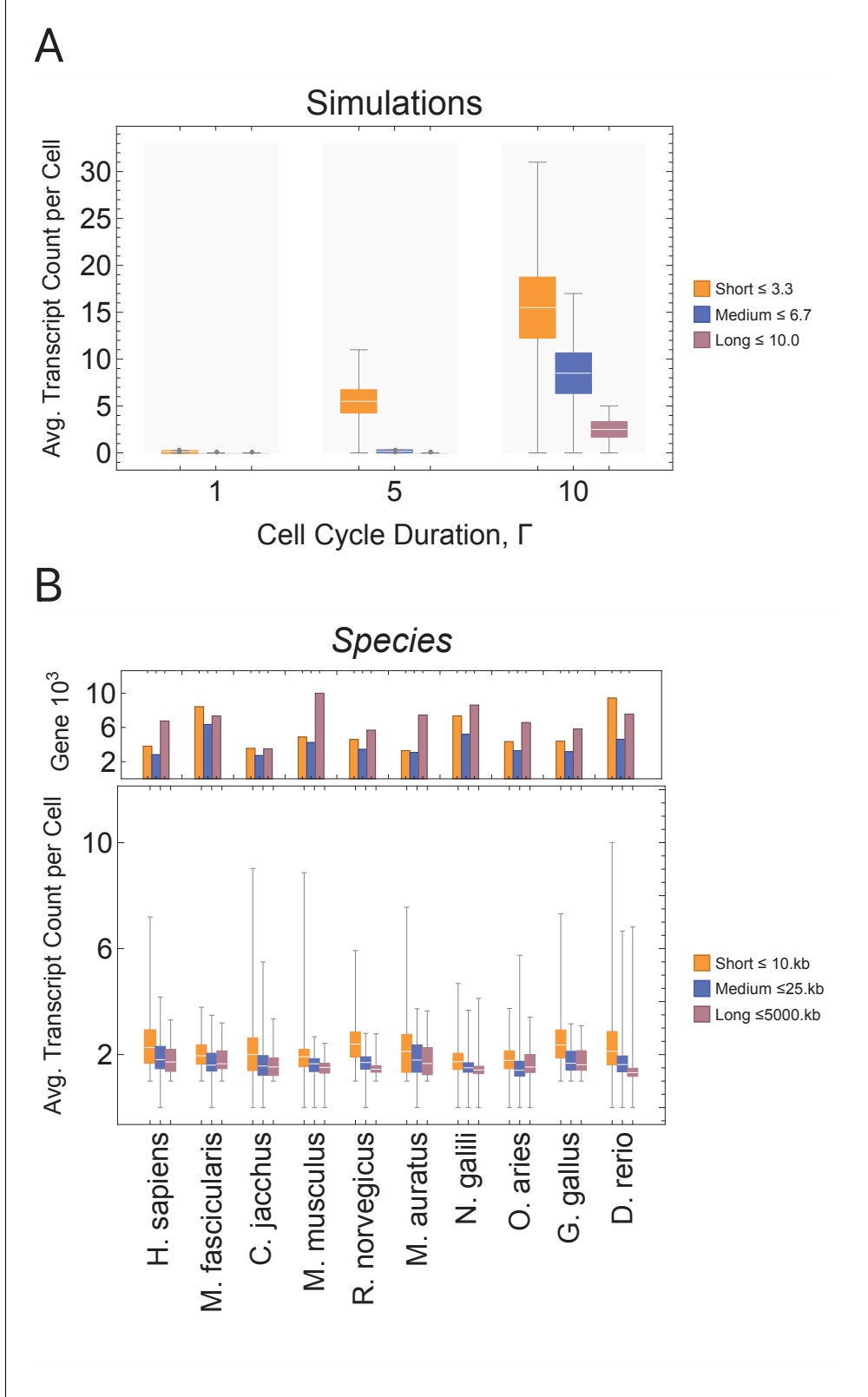

**Figure 3.** Short genes produce more transcripts than longer genes at multiple cell cycle duration lengths. The transcriptome for each cell is subdivided into short, medium, and long gene bins, and transcript counts are averaged per bin per cell. (**A**) Simulations predict that short gene transcripts will be more highly expressed than long gene transcripts, irrespective of the genome size. Simulation results are shown for cell cycle durations of 1, 5,

*Figure 3 continued on next page*

*Figure 3 continued*

and 10 hr and gene lengths (gene$^{L_1-L_{10}}$); see *Figure 3—figure supplement 3* for additional simulations (other parameters ploidy = 1, one cell division, iterations = 5,000,000, genome G = 10, gene$^{L_1-L_{10}}$, transcription rate, λ = 1 kb/hr, RNA polymerase II re-initiation, Ω = 0.25kb). Bins are defined such that genes are evenly distributed across them. (B) Single-cell microglia data obtained from GSE134707 (*Geirsdottir et al., 2019*) displaying expected patterns where short genes (lengths <10 kb) have a higher transcript expression than both medium genes (lengths > 10 kb) and longer genes (lengths >25 kb) – Kolmogorov–Smirnov test p < 10$^{-16}$, the upper bound p-value for all short-medium and short-long comparisons – across nine different species (age): *Macaca fascicularis* (3 years), *Callithrix jacchus* (7 years), *Mus musculus* (8–16 weeks), *Rattus norvegicus* (11–14 weeks), *Mesocricetus auratus* (8–16 weeks), *Nannospalax galili* (2-4 years), *Ovis aries* (18–20 months), *Gallus gallus* (24 weeks), and *Danio rerio* (4–5 months). The top part of the plot shows the total number of genes possible in each bin, given the gene length distribution of each genome. Bins are defined such that they are both consistent across all species and also approximately evenly filled with genes.

The online version of this article includes the following figure supplement(s) for figure 3:

**Figure supplement 1.** Simulations exploring the effects of cell cycle duration and RNA polymerase II (rnaPol II) for different re-initiation distances, Ω, and transcription rates, λ.

**Figure supplement 2.** Effects of maternal transcript inheritance.

**Figure supplement 3.** Simulations exploring the effects of cell cycle duration on transcript count per cell.

**Figure supplement 4.** Single-cell data exploring the effects of cell cycle duration on transcript count per cell.

---

simplicity, but these were repeated thousands of times to model cell population effects. We compute cell diversity in two ways; first, as the number of distinct transcriptomes in the cell population (transcriptome diversity); and second, as the number of distinct transcriptomic clusters, as defined using standard single-cell transcriptomic analysis techniques (*Satija et al., 2015*) (see Materials and methods). Both measures model real cell types and states that are distinguished by their transcriptomes, with transcriptome diversity as an upper bound on cell-type number, and cluster number approximating a lower bound. We first ran simulations with an active transcriptional filter by varying only the cell cycle duration, Γ, for a genome with 10 genes, with genes ranging in size from 1 to 10 kb, such that it satisfies $L_1 = 1 \leq ...\Gamma... \leq L_G = 10$. Short cell cycle duration parameter values generated a homogenous population of cells because only short transcripts can be transcribed. As cell cycle duration was increased, transcriptome diversity also increased. Longer cell cycle duration values generated heterogeneous populations because a range of transcripts can be expressed (*Figure 4A*, brown line). Interestingly, cell cluster diversity peaks at intermediate cell cycle duration parameter values (*Figure 4B*, brown line; *Figure 4C*) because new genes are introduced with increasing cell cycle lengths, but eventually long cell cycles provide sufficient time for cells to make all transcripts, which leads to reduced variance between the progeny. We next repeat this experiment by turning off the transcriptional filter by reducing the maximum gene length such that $L_G < \Gamma$ (*Figure 4A, B*, blue line). In this case, cell diversity can be generated, but it quickly saturates (*Figure 4B*, blue line), as all transcripts are expressed, given a cycle duration allowing the expression of the longest transcript. Thus, while cellular diversity can be generated with an active or inactive transcriptional filter, diversity is more easily controlled by cell cycle duration when the transcriptional filter is active.

In general, transcriptome diversity increases as a function of cell cycle duration (Γ), transcription rate (λ), and number of genes in the genome (G). In particular, transcriptome diversity = $n \prod_{i=1}^{G}(T/L_i + 1)$, where n is the genome ploidy level, $T = \sum_{a=0}^{\frac{L_i}{\Omega}-1} f(a), \forall f(a) \geq 0, f(a) = \Gamma * \lambda - \frac{a\Omega}{\lambda}$ (i.e., the maximum transcribed gene length, T, is restricted by the product of cell cycle duration, Γ, transcription rate, λ, and RNA polymerase II re-initiation, Ω), and $L_i$ is the length of gene$_i$. This analytical solution of cell transcriptome diversity was validated by comparing it to simulations (*Supplementary file 2*). While the number of genes and their length distribution can change over the course of evolution, these numbers are constant for a given species, and transcription rate is likely highly constrained (*Ardehali and Lis, 2009*), leaving only cell cycle duration as a controllable parameter of cell diversity during development, according to our model.

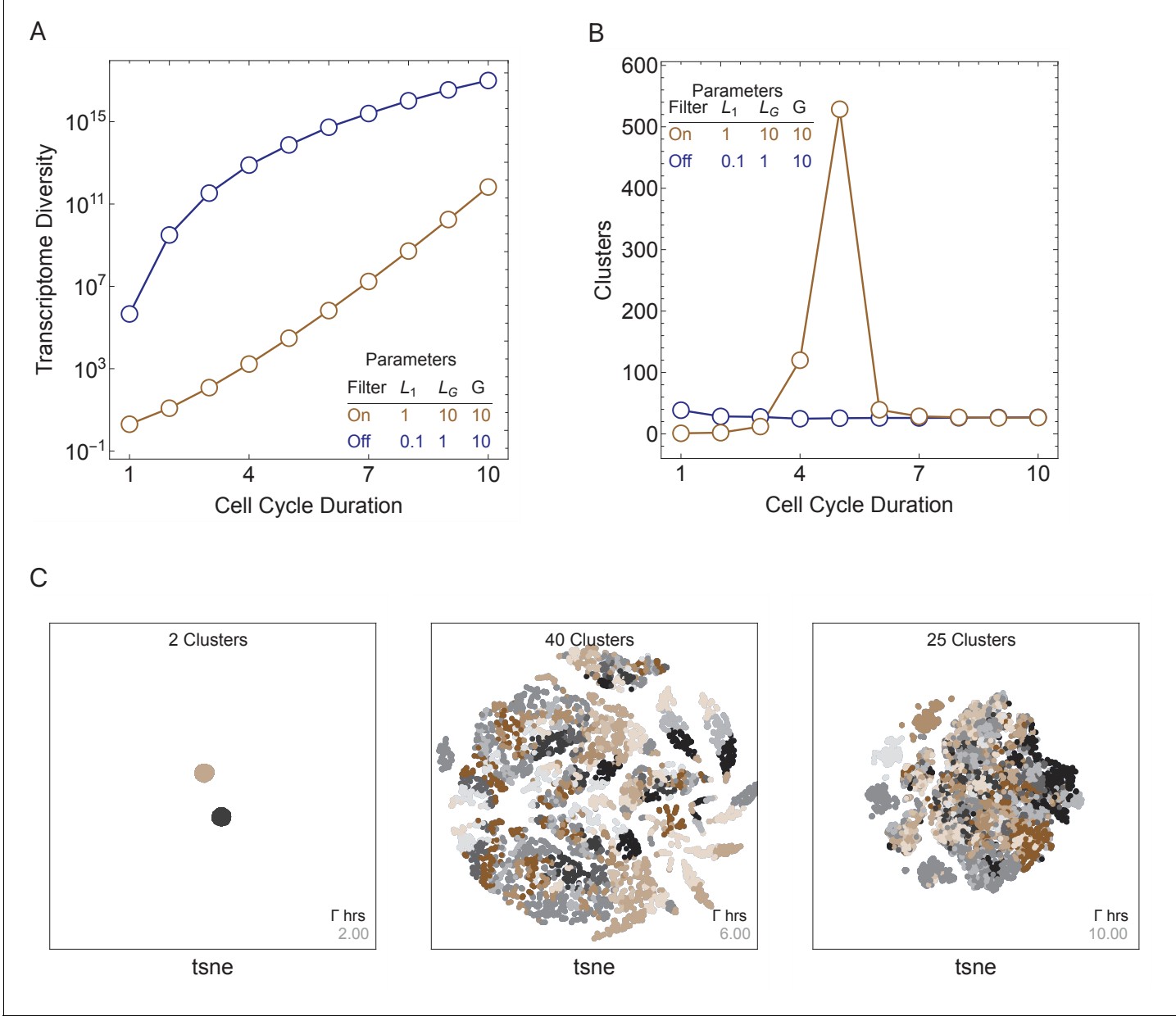

**Figure 4.** Cell cycle duration can control cell diversity. Simulations explore the effects of cell cycle duration, $\Gamma$, gene number, G, and gene length distribution. (A) Simulations show that cell diversity (transcriptome diversity) increases as a function of cell cycle duration. Short cell cycle durations can constrain the effects of gene number as long as a transcriptional filter is active (gene length distributions are broad, $L_1 < \ldots (\Gamma * \lambda) \ldots < L_G$). When $L_G < (\Gamma * \lambda)$, cell cycle duration does not control cell diversity. Cell cycle duration effects are relative to the gene length distribution in the genome. (B) We use Seurat to cluster the simulated single-cell transcriptomes (10,000 cells) using default parameters and report the number of cell clusters over the simulations. This shows that cell diversity increases with gene number, but the number of clusters identified decreases when all the gene transcripts can be expressed similarly among all cells. (C) Representative examples (10,000 cells) of t-SNE visualizations (RunTSNE using Seurat version 3.1.2) are shown for simulations with cell cycle durations 2, 6, and 10 hr (genome G = 10, gene$^{L-L}_{1\ 10}$, ploidy n = 1, and transcription rate, $\lambda$ = 1 kb/hr, RNA polymerase II re-initiation, $\Omega = 0.25\text{kb}$).

## Model prediction: varying cell cycle duration over developmental time controls tissue cell proportions and number

During multicellular organism development, it is essential to generate the correct numbers of cells and cell types. Cell cycle duration changes dramatically during development, generally starting out fast to generate cells quickly and slowing down over time as the organism matures

(*Supplementary file 1* and *Figure 1*; *Farrell and O'Farrell, 2014*; *O'Farrell et al., 2004*). Clearly, cells with short cell cycles generate more progeny compared to those with longer cell cycles. However, we propose that a tradeoff exists, balancing the generation of diversity (longer cell cycle durations) with the fast generation of cells (shorter cell cycle durations; *Figure 4B*). To study this tradeoff, we simulated cell propagation under a 'mixed lineage' scenario where, after the first division, one child cell and its progeny maintains a constant cell cycle duration ($\Gamma_1$ = 1 hr) and the second child cell and its progeny maintains an equal or longer constant cell cycle duration over a lineage with 20 cell division events (*Figure 5*, gray and blue lineages, respectively). We initialize the starting cell with no prior transcripts (naïve theoretical state) and a genome containing five genes ranging from length 1 to 2 kb (gene$^1$, gene$^{1.25}$, gene$^{1.5}$, gene$^{1.75}$, gene$^2$), setting cell cycle duration in the second lineage to range between 1 and 2, controlling the transcriptional filter threshold in this

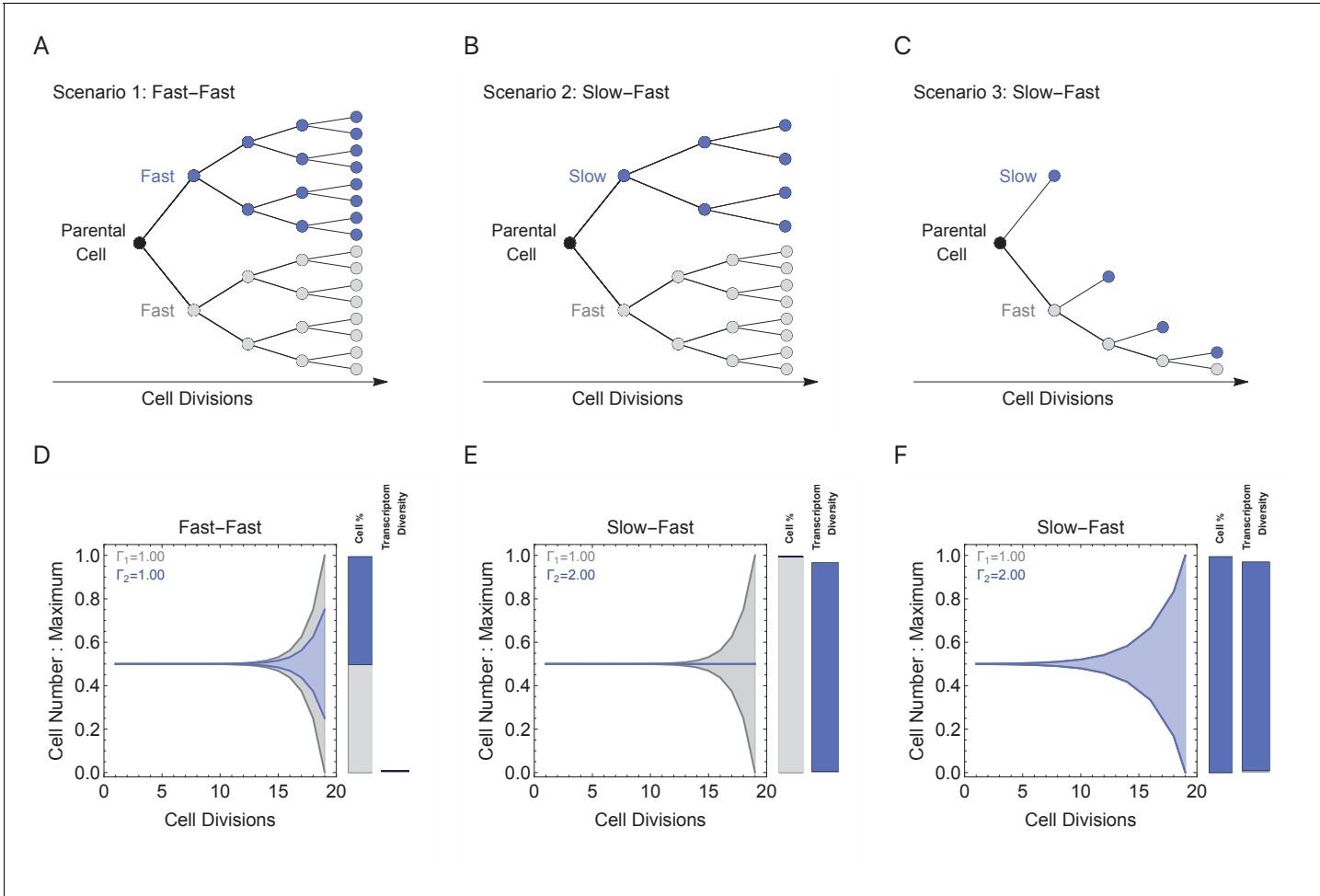

**Figure 5.** Cell cycle duration can control the generation of cell proportions and cell types within a population. Simulations start with two cells and run for 18 divisions (generating $2^{19}$ cells when cell cycles are the same). Cell$_1$ is initialized with cell cycle duration $\Gamma_1$ = 1 hr, Cell$_2$ has cell cycle duration, $\Gamma_2$, ranging from 1 to 2 hr. All progeny are tracked based on their cell cycle duration (lineage $\Gamma_1$ = 1 cell cycle duration, gray, or lineage $\Gamma_2$ cell cycle duration, blue). Tree plot depicting lineages when the cell cycle duration (**A**) is the same, $\Gamma_1 = \Gamma_2$ (scenario 1), or (**B, C**) differs, $\Gamma_1 < \Gamma_2$ (scenarios 2 and 3). Scenario 2 captures a situation when the cell cycle is determined by the parental lineage, while scenario 3 captures a situation when a cell splits asymmetrically into a fast and slow cell, resulting with the fast lineage having just one cell. (**D–F**) Müller visualizations show that when the cell cycle duration is the same, both cells contribute the same number of progeny and cell proportions (%) are 50:50 (bottom left panel). The visualization is stacked, down-scaling the blue lineage slightly to reduce occlusion of the gray lineage. Cells with longer cell cycle duration (blue lineage) generate fewer progeny with respect to the cells with a short cell cycle duration of 1 hr (gray lineage). However, the slower cells contribute more to the diversity observed in the population, shown as the blue and gray transcriptome diversity bars. Thus, increasing cell cycle duration increases cell diversity, but also limits the number of progeny generated. The system can overcome the limit on cell number by using scenario 3, where more slow cells can be generated (other parameters G = 5, gene lengths (gene$^{L-L}_{1\ 2}$), genome = {1,1.25,1.5,1.75,2} and ploidy = 1, RNA polymerase II re-initiation, $\Omega = 0.25$kb).

lineage only. We considered three scenarios: (1) both cell lineages cycle at the same rate (Fast-Fast, *Figure 5A*); (2) the first (blue) lineage is slower than the second (gray) (Slow-Fast, *Figure 5B*); and (3) both slow and fast lineages divide asymmetrically, producing one slow and one fast cell (Slow-Fast, *Figure 5C*).

In the simulation where both cell lineages cycle at the same rate (*Figure 5A*), both lineages generate the same number of progeny with the same level of diversity (*Figure 5D*). When cell cycle duration for the second (blue) lineage is increased across simulations (*Figure 5E*), the transcriptional filter acts to generate more diverse progeny, but with fewer cell numbers and progressively smaller population proportions due to the slower cell cycle (*Figure 5E*, blue bars). Meanwhile, the short cell cycle lineage maintains a steady, low level of diversity generation (*Figure 5E*, gray bars). When a fast cell can divide asymmetrically, generating one slow and one fast cell at each division, the number of slow cells in the population can increase; however, this comes with a reduction of the number of fast cells in the population (*Figure 5F*). Thus, our simulations show how the cell cycle duration parameter can impose a tradeoff between cell proportion and diversity generation, and mixing lineages with different cell cycle durations can generate mixed cell populations each with their own diversity levels.

To more faithfully simulate multicellular animal development where cell cycle duration increases over time, we next allowed progeny cells to differ in their cell cycle duration from their parents in each generation (*Figure 6A*). Increasing the cell cycle duration over time reveals that cell cycle dynamics can alter the number and proportions of cells as a function of time (cell generations; *Figure 6B* and *Figure 6—figure supplement 1*). To compare with a real system, we explore single-cell transcriptomics data measured over four time points of mouse cortex development (*Yuzwa et al., 2017*). Average cell cycle duration over mouse neural cortex development is known to increase from 8 hr at embryonic day 11 (E11) to an average of 18 hr by E17 (*Furutachi et al., 2015*; *Takahashi et al., 1995a*). Within this range, progenitor cells are, in general, expected to be characterized by fast cycles with short G1 duration and neurons by slower cell cycles with long G1 duration (*Calegari et al., 2005*). In our analysis of the mouse cortex scRNA-seq data, we find that genes with increasing transcript expression across the time course (E11.5 < E13.5 < E15.5 < E17.5) are associated with neural developmental (maturing cell) pathways, whereas the genes with decreasing transcript expression across time (E11.5 > E13.5 > E15.5 > E17.5) are associated with transcription and proliferation (stem and progenitor cell) pathways (*Figure 6—figure supplement 2*). Furthermore, we observe an overall pattern of an increasing number of cells with long cell cycle duration and a decrease in fast cycling cells (*Figure 6C*) following the same general trend as observed in our simulations (*Figure 6A*), supporting the idea that cell cycle duration dynamics could play a role in controlling cell proportions and cell diversity in a developing tissue.

## Hypothesis: a cell cycle-dependent transcriptional filter may help control cell proportion and diversity in tissue development

Our theoretical model and agreement with general trends in scRNA-seq data supports the hypothesis that a cell cycle-dependent transcriptional filter has the potential to control cell proportion and diversity in tissue development. In this section, we use the model to generate specific questions that can be checked in real data, further supporting our model.

## Organismal level

Our model suggests that organisms with long genes will need to maintain long cell cycle durations during development. Cell cycle duration measurements are not widely available, which makes directly testing this hypothesis difficult. Instead, we explored related questions. We started by asking if organisms with longer genes would also take longer to develop. We analyze gene length distributions for 12 genomes spanning budding yeast to human with a diverse range of developmental durations, as shown in *Figure 7* and *Supplementary file 3* (*Gilbert and Barresi, 2016*; *Jukam et al., 2017*). Non-mammalian species that we analyze are relatively fast developing, ranging from approximately 2 hr (e.g., *Saccharomyces cerevisiae*) to a few days (e.g., *X. tropicalis* and *D. rerio*), while mammals (*M. musculus*, *Sus scrofa*, *Macaca mulatta*, and *Home sapiens*) are relatively slow developing (20, 114, 168, and 280 days, respectively, *Supplementary file 3*). These species also have different gene length distributions; to illustrate this quantitatively, using a typical transcription rate of 1.5

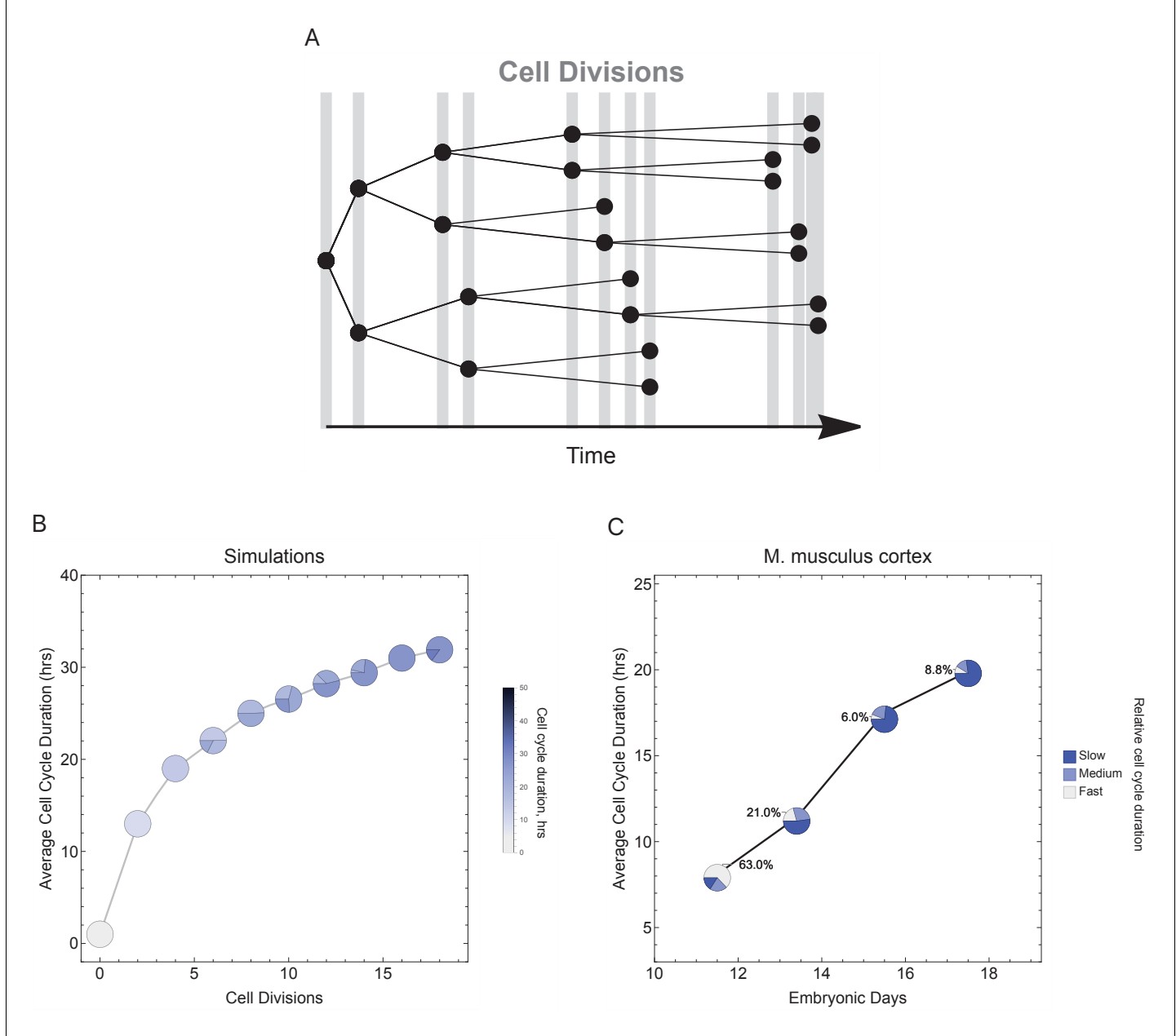

**Figure 6.** Varying cell cycle duration across time affects cell-type proportions. (**A**) Cell cycle duration increases after each cell division, with amount of increase defined using a Gaussian distribution. (**B**) Simulation of gradually increasing cell cycle duration over time, such that $\Gamma$ = Gaussian (mean $\Gamma_{parent}$ $\pm$ 6, standard deviation $\sigma$ = 0.06), affects the relative proportion of cells with different cell cycle durations (pie charts). All cell progeny are labeled based on their cell cycle duration (inherited from parent). See *Figure 6—figure supplement 1* for results using other increment rates. Parameters: genome = 10, gene lengths (gene$^{L-L}_{1\ 10}$), $\lambda$ = 1 kb/hr, 18 cell divisions, iterations = 500, ploidy n = 1, RNA polymerase II re-initiation, $\Omega = 0.25\mathrm{kb}$. (**C**) Single-cell transcriptomics data from GSE107122 (*Yuzwa et al., 2017*) for embryonic mouse cortex development, known to exhibit increasing cell cycle duration over time. This data includes identified cell types, is a time series, and we know the average cell cycle duration at each time point; at E11.5, the average cell cycle duration is 8 hr and by E17.5 it is 18 hr (*Furutachi et al., 2015*; *Takahashi et al., 1995a*). Cells were defined as relatively fast cycling cells (apical progenitors), relatively medium cycling (intermediate progenitors), and relatively slow cycling (neurons), with cell-type annotation based on cell clustering analyses conducted in (*Yuzwa et al., 2017*). We show how cell proportions (pie charts) change across time, with apical progenitors (relatively fast cycling cells) decreasing in frequency as the average cell cycle duration increases.

The online version of this article includes the following figure supplement(s) for figure 6:

**Figure supplement 1.** Varying cell cycle duration across time affects cell-type proportions.

**Figure supplement 2.** Genes with increasing transcript expression are associated with neuronal and synaptic pathways.

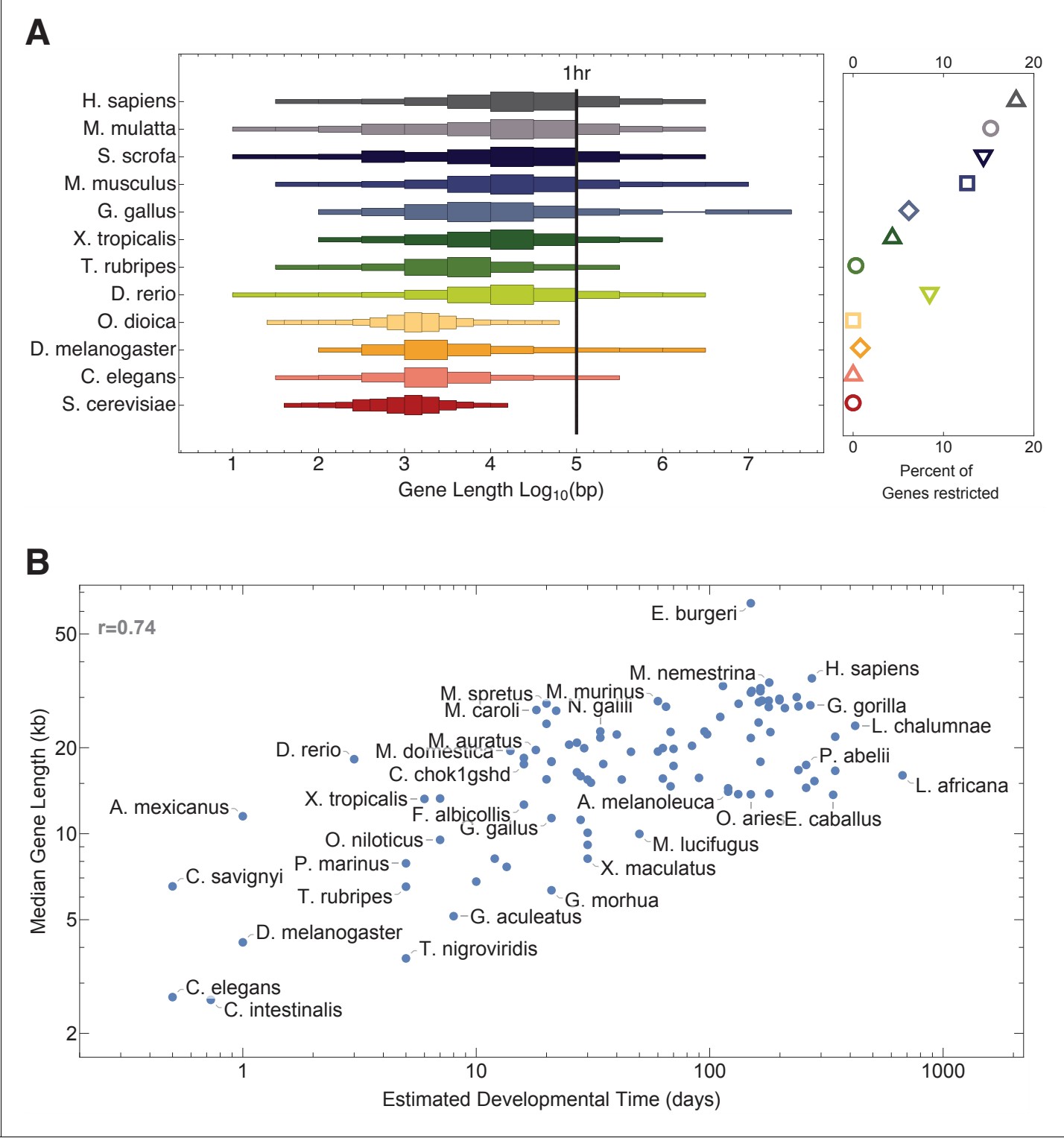

**Figure 7.** Gene length distribution and developmental time are correlated. (**A**) Model organisms exhibit a large diversity in gene length distributions over their genomes. Species that have narrower gene length distributions tend to develop faster, while slow developers (mammals) exhibit broad and right-shifted gene length distributions. Demarcating a 1 hr cell cycle duration using an average transcription rate of 1.5 kb/min illustrates the proportion of genes that would be interrupted before transcript completion for each organism. *Saccharomyces cerevisiae* (budding yeast), *Caenorhabditis elegans* (worm), *Drosophila melanogaster* (fruit fly), *Oikopleura dioica* (tunicate), *Danio rerio* (zebrafish), *Takifugu rubripes* (fugu), *Xenopus tropicalis* (frog), *Gallus gallus* (chicken), *Mus musculus* (mouse), *Sus scrofa* (pig), *Macaca mulatta* (monkey), and *Homo sapiens* (human). (**B**) There is a clear positive correlation

*Figure 7 continued on next page*

*Figure 7 continued*

between developmental time and median gene length (101 species, *Figure 7—figure supplement 1*). Estimated developmental time was curated from the Encyclopedia of Life or articles found in PubMed (*Supplementary file 3*). We used gestation time for mammals and hatching time for species who lay eggs (since it is difficult to accurately define a comparative stage for all species). We analyzed the data using a Pearson correlation test, shown as r. For each species, we calculated median gene length: all protein coding genes were downloaded from Ensembl version 95 (*Yates et al., 2016*) using the R Biomart package (*Durinck et al., 2009*; *Durinck et al., 2005*). The length of each gene was calculated using start_position and end_position for each gene as extracted from Ensembl data.

The online version of this article includes the following figure supplement(s) for figure 7:

**Figure supplement 1.** Illustration of the association between developmental time and median gene length across 101 species, grouped by taxonomy class (*Supplementary file 3*).

kb/min (*Ardehali and Lis, 2009*), a cell cycle duration of 1 hr can exclude up to 20% total genes found in relatively slow developers and not exclude any genes in fast developers (*Figure 7A*). In agreement with our hypothesis, the gene length distribution is narrower and left shifted (shorter genes) for fast developers and broader and right shifted (longer genes) for slower developing species. Interestingly, one seeming exception to the overall gene length distribution trend in multicellular animals is the tunicate *Oikopleura dioica*, which has relatively short genes, but also has a rapid gestation period of 4 hr to hatched tadpole (approximately twice as fast as *C. elegans* and six times faster than *D. melanogaster*), supporting our analysis. Broadening this analysis to 101 species, we again find an association (r = 0.74) between estimated developmental time and median gene length (*Figure 7B* and *Figure 7—figure supplement 1*).

Our model suggests that short genes will be enriched in pathways that can function independently from long genes, and that long genes may be enriched in pathways related to mature, differentiated cell types with slower cell cycles (*Figure 8B*). We examined the functions of short and long genes by conducting a pathway enrichment analysis on all genes in a genome ranked by their length. In the human genome, the longest genes are enriched in processes such as neural development, muscle control, cytoskeleton, cell polarity, and extracellular matrix, and the shortest genes are enriched in processes that presumably need to be quickly activated transcriptionally (e.g., immune, translation, and environment sensing; *Figure 8—figure supplement 1*). We performed a similar pathway analysis for human (*Figure 8* and *Figure 8—figure supplements 2* and *3*) and 12 other species (*Figure 9*) and found general agreement with these patterns, finding the longest genes (gene length in the 95% quantile) enriched for genes involved in mature cell-related processes (e.g., brain and muscle development), whereas the shortest genes (gene length in the 5% quantile) are enriched for genes involved in core processes (e.g., immune, RNA processing, and olfactory receptors).

## Spatial level

Within an organism, cell cycle duration and transcript expression vary across many factors, including spatially. We hypothesize that spatial transcript expression patterns can be initially organized by gene length. To explore this, we study the developing fruit fly embryo (*D. melanogaster*) where the average cell cycle rates differ spatially (*Foe, 1989*). At the onset of cell cycle 14, cells in different embryo regions start to divide at different rates, caused by an increase in their gap phase length, varying from 30 min to 170 min (*Foe, 1989*; *Foe and Alberts, 1983*). Cell cycle duration lengthening is spatially organized, with anterior regions dividing faster than posterior regions, with the mid-ventral region being the slowest (*Figure 10*). The embryo also exhibits spatial segregation patterns due to Hoxd gene family transcript expression (*Mallo and Alonso, 2013*). Overlaying the spatial patterns of hox gene family transcript expression and cell cycle duration obtained from independent studies, we observe that fast cycling regions express the shortest hox genes (Dfd 10.6 kb, lab 17.2 kb) and slow cycling regions express the longest hox genes (Ubx 77.8 kb and Antp 103.0 kb) (*Foe, 1989*; *Lemons and McGinnis, 2006*) in agreement with our model.

## Discussion

How cellular processes support the carefully orchestrated timing of tissue development that results in a viable multicellular organism is still unclear. While a combination of many potential cell autonomous and non-autonomous mechanisms, such as cytoplasmic molecules and gradients, cell-cell

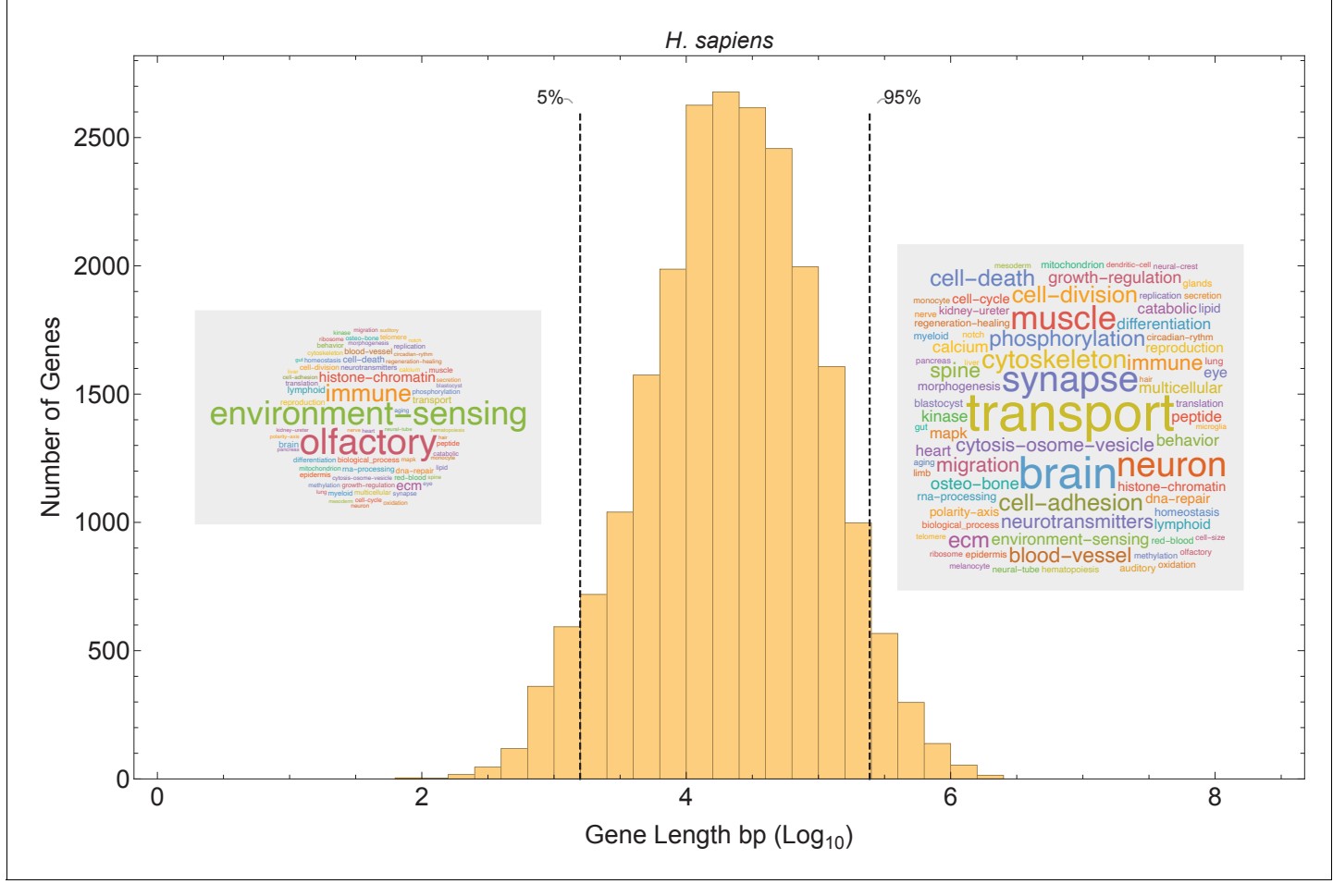

**Figure 8.** Short genes and long genes participate in different pathways. The plot shows the *H. sapiens* gene length distribution. We selected the shortest 5% quantile as a list of short genes and the 95% quantile as a list of long genes. Short genes < 1.6 kb (n=1124) are involved in immune defense, environment-sensing, and olfactory, and long genes >243 kb (n = 1125) are represented in processes involving muscle and brain development, as well as morphogenesis. For each gene group, we identified all corresponding Gene Ontology (*Ashburner et al., 2000*) biological process terms downloaded from the Ensembl genome database version 100 (*Yates et al., 2016*), grouped the terms into themes (*Supplementary files 5* and *6*), and visualized the resulting term frequencies as word clouds using Mathematica. Refer to *Figure 8—figure supplements 2* and *3* for a more detailed analysis of the themes across all gene groups.

The online version of this article includes the following figure supplement(s) for figure 8:

**Figure supplement 1.** Enriched pathways in short (**A**) and long (**B**) genes in human.

**Figure supplement 2.** Genes participate in different pathways.

**Figure supplement 3.** Moving average across gene length.

**Figure supplement 4.** Pathway themes are associated with gene length.

communication, microenvironment signals, and effective cell size (*Edgar et al., 1986*; *Mukherjee et al., 2020*; *Tabansky et al., 2013*; *Yoon et al., 2017*), are likely important, one hypothesis is that gene length can be used as a mechanism to control transcription time in this process (*Artieri and Fraser, 2014*; *Gubb, 1986*; *Keane and Seoighe, 2016*; *Swinburne et al., 2008*). Bryant and Gardiner further hypothesize that cell cycle duration may play a role in filtering genes that influence pattern formation and regeneration (*Bryant and Gardiner, 2018*; *Ohsugi et al., 1997*) as cell cycle lengthens over development (*Figure 1* and *Supplementary file 1*; *Foe, 1989*; *Foe and Alberts, 1983*; *Newport and Kirschner, 1982b*; *Takahashi et al., 1995b*). Early experiments using embryos suggested that cell cycle duration has a role in transcription initiation; however, these experiments lacked the temporal resolution necessary to dissociate the effects of cell cycle duration and transcriptional control from other mechanisms (*Edgar et al., 1986*; *Edgar et al.,*

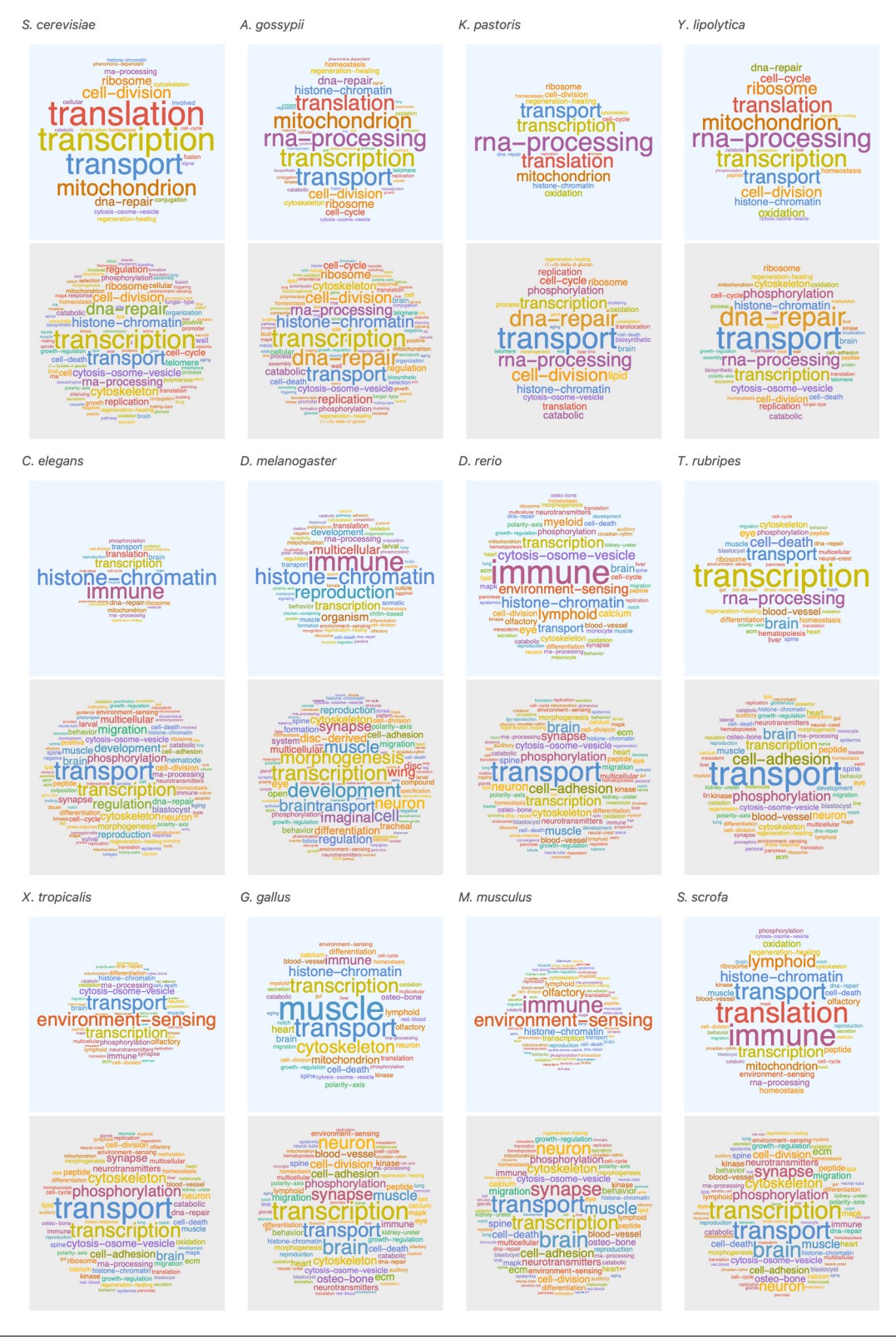

**Figure 9.** Short genes exhibit different pathways than long genes, and this trend is consistent across a wide species range. We selected the shortest 5% quantile as a list of short genes (top panels in blue) and genes above the 95% quantile to define a list of long genes (bottom panels in gray). *Saccharomyces cerevisiae* (short < 0.24 kb, long > 3.5 kb), *Ashbya gossypii* (short < 0.36 kb, long > 3.5 kb), *Komagataella pastoris* (short < 0.37 kb, long > 3.3kb), *Yarrowia lipolytica* (short < 0.39 kb, long > 3.5 kb), *Caenorhabditis elegans* (short < 0.47 kb, long > 9.6 kb), *Drosophila melanogaster*
*Figure 9 continued on next page*

*Figure 9 continued*

(short < 0.56 kb, long > 29 kb), *Danio rerio* (short < 1.3 kb, long > 127 kb), *Takifugu rubripes* (short < 0.72 kb, long > 27 kb), *Xenopus tropicalis* (short < 0.93 kb, long > 83 kb), *Gallus gallus* (short < 0.67 kb, long > 104 kb), *Mus musculus* (short < 1.2 kb, long > 183 kb), and *Sus scrofa* (short < 0.57 kb, long > 197 kb). For each gene group, we identified all corresponding Gene Ontology biological process terms from the Ensembl genome database (100) and visualized the resulting term frequencies as word clouds using Mathematica.

*1994*; *Kimelman et al., 1987*; *Newport and Kirschner, 1982b*; *Newport and Kirschner, 1982a*). It is also well known that cell cycle length changes can control cell fate and development (*Coronado et al., 2013*; *Mummery et al., 1987*; *Pauklin and Vallier, 2013*; *Singh et al., 2013*); however, this has remained observational and not linked to a mechanism. To help address these limitations, we developed an in silico cell growth model to directly study the relationship between cell cycle duration and gene transcription in a developmental context. The new discovery we make is that a transcriptional filter can be controlled by cell cycle duration and used to simultaneously control the generation of cell diversity, the overall cell growth rate, and cellular proportions during development (defining an emergent property of our computational model – see Appendix 1). Genomic information (gene number and gene length distribution) and cell cycle duration are critical parameters in this model. Across evolutionary time scales, cell diversity can be achieved by altering gene length (*Keane and Seoighe, 2016*); however, in terms of developmental time scales, we propose that cell cycle duration is an important factor that may control cell diversity and proportions within a tissue.

We predict that increasing the gene length distribution across a genome over evolution can provide more cell cycle-dependent transcriptional control in a developing system, leading to increased

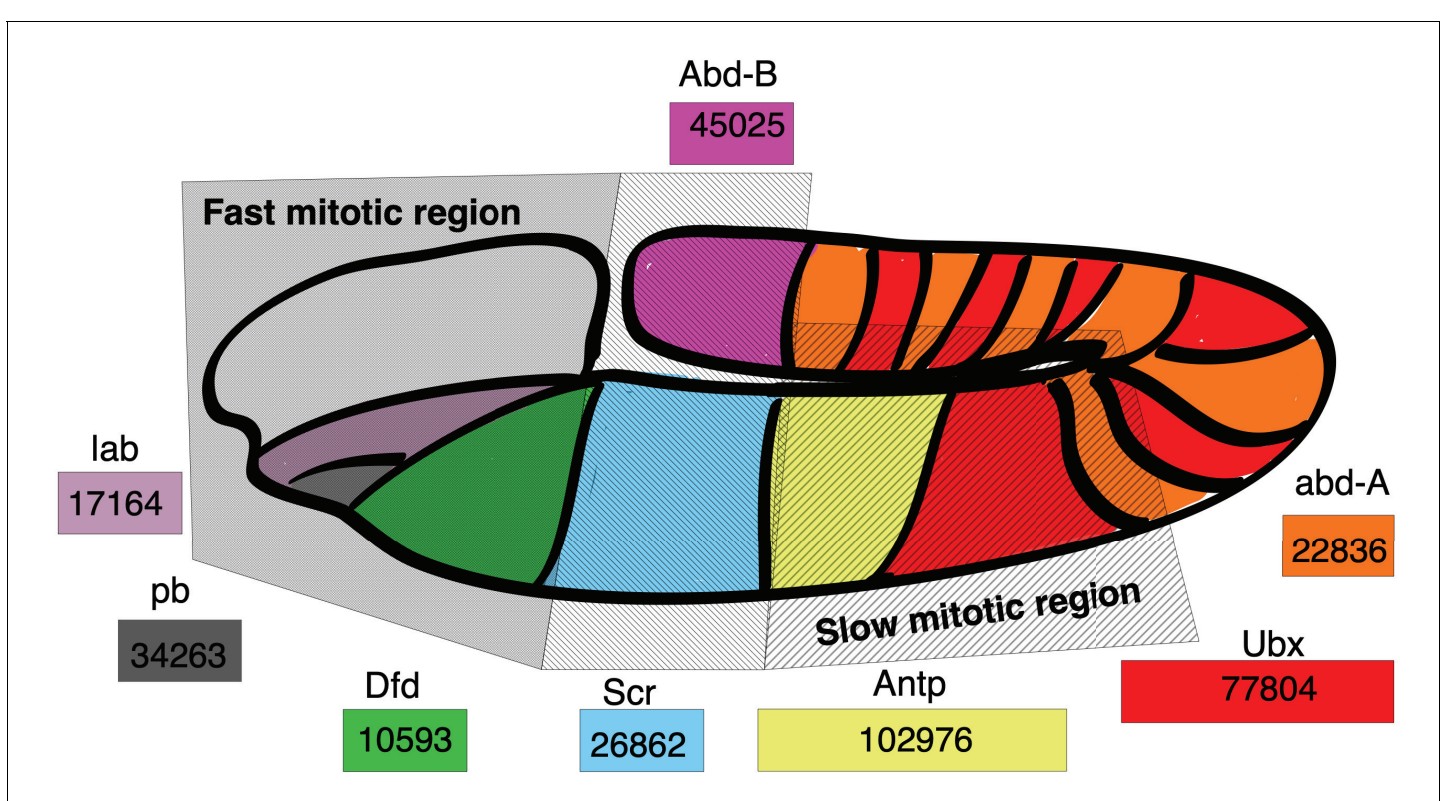

**Figure 10.** Hox gene length is correlated with spatial expression and cell cycle duration in the *D. melanogaster* embryo. *Drosophila* Hoxd family genes are each represented by a colored rectangle, containing the length of the gene in base pairs. Spatial expression of a gene transcript is marked by its corresponding color on the *Drosophila* embryo map. Hoxd gene length is correlated with the cell cycle duration of the embryo location where the gene transcript is expressed, with short Hox gene transcripts expressed in regions with short mitotic cycles and long Hox gene transcripts expressed in regions of long mitotic cycles. Spatial map of cell cycle duration from *Foe, 1989*; *Foe and Alberts, 1983* and gene transcript expression from *Mallo and Alonso, 2013*.

cellular diversity. Examining a range of genomes and associated data provides support for this novel idea. We observe that fast-developing organisms have shorter median gene lengths relative to the broad distributions, including many long genes, exhibited by slow developers (mammals). This aspect of genome structure may help explain the observed rates of cell diversity and organism complexity, as measured by number of different cell types, over a wide range of species (*Figure 7—figure supplement 1*; *Valentine et al., 1994*; *Vogel and Chothia, 2006*).

While we hypothesize that a cell cycle-dependent transcriptional filter is a fundamental regulatory mechanism operating during development (because gene length is fixed in the genome and transcription rate is expected to lie in a narrow range), multiple other regulatory mechanisms could modulate its effects. Furthermore, exploring these mechanisms may even result in similar conclusions as it can be evolutionary advantageous to have multiple paths to the same outcome; these include, but are not limited to, silencing or deactivating genes, gene regulatory networks, blocking gene clusters, for example, Hoxd (*Rodríguez-Carballo et al., 2019*) changing the transcription or re-initiation rate of RNA polymerase II (*Figure 3—figure supplement 1*), or inheriting long transcripts maternally at the zygote stage (*Figure 3—figure supplement 2*). Our current model only explores the effects of transcription and re-initiation rates of RNA polymerase II, mRNA transcript degradation rates, and maternally introduced transcripts. For the latter mechanism, we expect longer transcripts to be major contributors during the early maternal phase (*Jukam et al., 2017*), which agrees with zebrafish (*D. rerio*) experiments showing that maternal transcripts are longer and have evolutionary conserved functions (*Heyn et al., 2014*). Indeed, if we add maternal transcript inheritance to our model, we see the same pattern of a small number of long transcripts present early, as expected (*Figure 3—figure supplement 2*). Future work would entail curating experimental data about more regulatory mechanisms in cell systems and testing their association with cell cycle duration.

Our analysis raises interesting directions for future work. We focus on development, but transcriptional filtering may be important in any process involving cell cycle dynamics, such as regeneration (*Bryant and Gardiner, 2018*), wound repair, immune activation, and cancer. We must also more carefully consider cell cycle phase as transcription mainly occurs in the gap phases (*Bertoli et al., 2013*; *Newport and Kirschner, 1982b*). Experiments indicate that a cell will have different fates depending on its phase (*Dalton, 2013*; *Pauklin and Vallier, 2013*; *Vallier, 2015*). This agrees with our model as a cell at the start of its cell cycle will have a different transcriptome in comparison to the end of the cell cycle. Induced pluripotent cell state is also associated with cell cycle phases (*Dalton, 2015*), and efficient reprogramming is only seen in cell subsets with fast cell cycles (*Guo et al., 2014*). Our model could explain these observations as slower cycling cells could express long genes that push a cell to differentiate rather than reprogram. However, our model is limited to total transcription duration for interphase (G1, S, and G2), thus a future direction would be to explore different durations for each cell cycle phase. Collecting more experimental data about cell phase in developing systems will help explore these effects. Further, it will be important to explore how cell cycle duration is controlled. Molecular mechanisms of cell cycle and cell size (*Liu et al., 2018*) control could be added to our model to provide a more biochemically realistic perspective on this topic. Ultimately, a better appreciation of the effects of cell cycle dynamics will help improve our understanding of a cell's decision-making process during differentiation and may prove useful for the advancement of tools to control development, regeneration, and cancer. Finally, it is important to note that we have not provided experimental model support, only analyses that do not disagree with model predictions. We have also not proven the generality of the results across species. However, we hope that the hypotheses we explore here motivate new experimental studies to directly test the validity and generality of our model.

## Materials and methods

**Key resources table**

| Reagent type (species) or resource | Designation | Source or reference | Identifiers | Additional information |
|---|---|---|---|---|

*Continued on next page*

*Continued*

| Reagent type (species) or resource | Designation | Source or reference | Identifiers | Additional information |
|---|---|---|---|---|
| Software, algorithm | | *Wolfram, 2017* | | Mathematica (Wolfram Research Inc, Mathematica Versions 11.0–12, Champlain, IL, 2017) http://www.wolfram.com/mathematica/ |
| Software, algorithm | | This paper | | Cell developmental model https://github.com/BaderLab/Cell_Cycle_Theory |
| Software, algorithm | | PMID:25867923 *Satija et al., 2015* | | Seurat (3.1.2) https://satijalab.org/seurat/ |
| Software, algorithm | | PMID:26687719 *Yates et al., 2016* | | Ensembl (95) and (100) https://useast.ensembl.org/index.html |
| Software, algorithm | | PMID:10802651 *Ashburner et al., 2000* | | Gene Ontology http://geneontology.org/ |
| Software, algorithm | | PMID:16082012 *Durinck et al., 2005* | | BioMart (3.10) http://useast.ensembl.org/biomart/martview/ |
| Software, algorithm | | PMID:21085593 *Merico et al., 2010* | | Enrichment Map software (3.3.0) https://www.baderlab.org/Software/EnrichmentMap |
| Software, algorithm | | PMID:14597658 *Kucera et al., 2016* | | AutoAnnotate App https://baderlab.org/Software/AutoAnnotate |
| Software, algorithm | | PMID:14597658 *Shannon et al., 2003* | | Cytoscape (3.8.0) https://cytoscape.org/ |
| Software, algorithm | | PMID:30664679 *Reimand et al., 2019* | | Baderlab pathway resource (updated June 1, 2020) http://download.baderlab.org/EM_Genesets/ |

## Mathematical model

Our mathematical model is agent and rule-based. A single cell behaves and interacts according to a fixed set of rules. Our major rule involves a gene length mechanism, where each cell is defined by a genome and a cell cycle duration. The cell cycle duration determines which gene transcripts are expressed within the cell, based on the transcription rate. All decisions are based on a cell's autonomous information, and we omit external factors. We deliberately choose to consider this simple baseline setup to clarify the contribution of cell cycle duration to overall cell population growth.

Each cell is defined by a genome G (containing a set of genes), cell cycle duration in hours, and the transcripts inherited or recently transcribed. In the genome, each gene is defined by a length, gene$^{Length}$. For example, in a genome with three genes, (gene$^1$, gene$^2$, gene$^3$) represents genes of length 1, 2, and 3 kb, respectively.

Each cell can divide and make two progeny cells. This process can continue many times to simulate the growth of a cell population, and we keep track of the entire simulated cell lineage. For each cell division (one time step in the simulation): each Cell$_i$ will transcribe its genes based on the time available, defined by the cell cycle duration. We assume that the time it takes to transcribe a gene depends on its length and a fixed transcription rate; although a simplification, there are examples where this occurs, for instance, the human dystrophin gene is 2,241,765 bp long and takes about 16 hr to transcribe (*Tennyson et al., 1995*). Once a cell cycle is finished, the cell divides. When cells are synchronized, the first cell division T = $\Gamma_i$. When the cells are asynchronized, then the algorithm identifies the time allocated as the shortest cell cycle duration in the population as the time step and each cell division will have a different duration. In this case, we keep track of the exact duration such

that cells with short cell cycles, for example, $\Gamma$ = 1 hr, will register 10 divisions in 10 hr while cells with long cell cycles, for example, $\Gamma$ = 10 hr, will register one division in over the same time. We limited the model to two modes of division, symmetric (where the cell gives rise to identical cells, e.g., *Figure 5A*) and asymmetric (where the cell gives rise to a fast and slow cell, e.g., *Figure 5C*). We do not consider mechanisms that reduce cell numbers (cell death). For certain experiments (e.g., *Figure 6*), the cell cycle duration for each progeny is allowed to diverge from the parental duration using a monotonic function (increasing or decreasing) and a stochastic variable based on a Gaussian distribution with a mean equal to $\Gamma_i$ (parental cell cycle duration). This models a more realistic noisy distribution of cell cycle durations in the simulated cell population. The cell cycle and division rules are repeated for all cells in the population until a set number of cell divisions have been reached.

During a cycle, each cell contains a certain number of transcripts. The number of transcripts for each gene is calculated by a function of cell cycle duration, $\Gamma$, transcription rate, $\lambda$, re-initiation distance, $\Omega$, and gene length, $L: \sum_{a=0}^{\frac{\text{gene}^{L_i}}{\Omega}-1} \Gamma * \lambda - \frac{a\Omega}{\lambda}$. If the cell does not divide, then the number of transcripts reflects the current cell cycle phase, which is computed and stored. If the cell can divide within the time $T = (\Gamma * \lambda)$, then it will randomly, according to a uniform distribution, assign its transcripts between its two progeny cells. Typically, simulations were conducted with $\lambda$=1, simplifying the analysis to $(\Gamma - a\Omega)/\text{gene}^{L_i}$; however, we also explored the effects of transcript re-initiation and transcription rate on the system as shown in *Figure 3—figure supplement 1*.

Our model tracks single cells, with each cell identified by a transcriptome and cell cycle duration. The transcriptome data resemble a scRNA-seq matrix to aid comparison between simulation and experimental data. We allow cells without any transcripts, for example, (0,0,0) to exist – due to the low numbers of genes considered in our simplified model and results, and that parental transcripts are distributed between progeny, there is a probability of 2/(the total number of transcripts) that all the transcripts will end up in only one of the new cells, leaving the other one empty (*Zhou et al., 2011*). Theoretically we have no reason to omit these cells, and they may represent the most naïve theoretical state of a cell without any prior information. Early embryos, such as in *Xenopus* stages that lack zygotic transcription, may be similar real systems to such a state (*Newport and Kirschner, 1982b*).

Parameters tracked for each cell$_i$ = (number of divisions, current cell cycle phase, current time in cell cycle, length until next division, relative time passed, total cell cycle duration, transcriptome list, cell name, and lineage history). All cells are set with the same genome, ploidy level, and RNA polymerase II transcription rate and RNA polymerase II re-initiation distance.

Our model was developed and simulated using Mathematica (*Wolfram, 2017*).

## Quantification and statistical analysis

### Gene length analysis

All protein coding genes were downloaded from Ensembl genome database version 95 or 100 (*Yates et al., 2016*) using the R (3.6.1) Biomart package version 3.10 (*Durinck et al., 2005*). The length of each gene was calculated using start_position and end_position for each gene, as extracted from the Ensembl database (*Yates et al., 2016*).

## Single-cell analysis pipeline

Simulated data sets were preprocessed and clustered in R using the standard workflow implemented in the Seurat package version 3.1.2 (*Satija et al., 2015*). We used default parameters unless otherwise stated. Data were log-normalized and scaled before principal component analysis (PCA) was used to reduce the dimensionality of each data set. Due to the small number of simulated genes in our experiments, the maximum number of PCs (one fewer than the number of genes dims = 1:3) was calculated and used in clustering. FindVariableFeatures was used with loess.span set to 0.3 unless the number of genes was less than 5, then (0.4, 0.7, and 1 were used for simulations with 4, 3, and 2 genes, respectively). Cells were clustered using a shared nearest neighbor (SNN)-based 'Louvain' algorithm implemented in Seurat with reduction set as 'pca.' The clustering resolution was set to 1 for all experiments, and all calculated PCs were used in the downstream clustering process using the Louvain algorithm accessed via Seurat. Data was visualized with t-SNE after clustering.

## Developmental time curation

Estimated developmental time was curated from the Encyclopedia of Life or PubMed accessible articles (*Supplementary file 3*). We used gestation time for mammals and hatching time for species who lay eggs (since it is difficult to accurately define a comparative stage for all species). Species were grouped based on their taxonomic class and their developmental time was estimated by calculating the average number of days from zygote to birth or hatching.

## Pathway enrichment analysis

We used Gene Set Enrichment Algorithm (GSEA version 4.0.2), in pre-ranked analysis mode, to identify pathways enriched among all genes in a genome ranked by gene length (*Subramanian et al., 2005*). Gene ranks started at (number of genes)/2 to its negative equivalent and were normalized such that we generated a ranked list from 1 to −1, with 1 specifying the shortest gene and −1 the longest. The ranked gene length list was analyzed for pathway enrichment GSEA with parameters set to 1000 gene set permutations and gene set size between 15 and 200. Pathways used for the analysis were from Gene Ontology biological process (*Ashburner et al., 2000*), MSigDB c2 (*Ashburner et al., 2000*), WikiPathways (*Slenter et al., 2017*), Panther (*Mi, 2004*), Reactome (*Croft et al., 2011*), NetPath (*Kandasamy et al., 2010*), and Pathway Interaction database (*Schaefer et al., 2009*) downloaded from the Bader lab pathway resource (http://baderlab.org/GeneSets). An enrichment map, created using the EnrichmentMap Cytoscape app version 3.3.0 (*Merico et al., 2010*), was generated using Cytoscape (version 3.8.0) using only enriched pathways with p-value of 0.05 and FDR threshold of 0.01 (*Reimand et al., 2019*). Cross-talk (shared genes) between pathways was filtered by Jaccard similarity greater than 0.25. Pathways were automatically summarized using the AutoAnnotate App to assign pathways to themes (*Kucera et al., 2016*). Themes were further summarized by grouping pathways into more general themes with a mixture of automatic classification using key words and manual identification.

## Pathway word cloud analysis

All Gene Ontology pathways (GO biological processes) were downloaded from the Ensembl genome database, version 100 (*Yates et al., 2016*), using the R Biomart package version 3.5 (*Durinck et al., 2005*). We restricted analysis to pathways with at least three genes. We grouped genes based on their gene length (see *Gene length analysis* for details) and identified the pathways associated with each gene. The description of each pathway was collected and the frequency of each word within the pathway name was calculated. We defined themes (*Supplementary files 5* and *6*) for all *H. sapiens* available pathways (using only GO biological processes). Common, generic, and uniformly distributed themes (such as cellular response, metabolic biosynthesis, protein processes, signaling, and transcription) were manually removed from the list. The frequencies were visualized as word clouds using Mathematica (*Wolfram, 2017*).

## Data and code availability

Our simulation code is available at https://github.com/BaderLab/Cell_Cycle_Theory (*Chakra, 2021* copy archived at swh:1:rev:7eb38b679e917ba8522b17edae5498990a221ffc).

# Acknowledgements

We thank our reviewers for insightful comments. We thank Zain Patel, Brendan Innes, Derek van der Kooy, Peter Zandstra, Nika Shakiba, Janet Rossant, Eszter Posfai, Maria Shutova, Andras Nagy, and Rudy Winklbauer for thoughtful discussions about this work. This work was funded by the University of Toronto Medicine by Design initiative, by the Canada First Research Excellence Fund.

# Additional information

## Funding

| Funder | Author |
| --- | --- |
| Canada First Research Excel- | Gary D Bader |

lence Fund

| University of Toronto | Gary D Bader |

The funders had no role in study design, data collection and interpretation, or the decision to submit the work for publication.

### Author contributions

Maria Abou Chakra, Conceptualization, Data curation, Formal analysis, Validation, Investigation, Visualization, Methodology, Writing - original draft, Writing - review and editing, Development of the cell model; Ruth Isserlin, Data curation, Formal analysis, Pathway analysis using GSEA, cytoscape and enrichment map; Thinh N Tran, Formal analysis, Single cell analysis of mouse, xenopus and zebrafish data; Gary D Bader, Conceptualization, Formal analysis, Supervision, Funding acquisition, Validation, Investigation, Visualization, Writing - original draft, Writing - review and editing

### Author ORCIDs

Maria Abou Chakra ⓘ http://orcid.org/0000-0002-4895-954X

Gary D Bader ⓘ https://orcid.org/0000-0003-0185-8861

### Decision letter and Author response

Decision letter https://doi.org/10.7554/eLife.64951.sa1
Author response https://doi.org/10.7554/eLife.64951.sa2

# Additional files

### Supplementary files

- Supplementary file 1. Curated cell cycle duration data.

- Supplementary file 2. Simulations supporting transcriptome diversity analytical solution.

- Supplementary file 3. Curated developmental time for species and their corresponding median gene length. The length of each gene was calculated using start and end positions for each gene as extracted from the Ensembl genome database (version 95). Estimated developmental time was curated from the Encyclopedia of Life or articles found in PubMed.

- Supplementary file 4. General pathway themes from *Figure 8—figure supplement 1* generated by using a mixture of automatic classification applying key words and manual identification.

- Supplementary file 5. General pathway themes and their corresponding list of words that were used to manually classify the pathways in *Figure 8* and *Figure 8—figure supplements 2* and *3*.

- Supplementary file 6. General pathway themes in *Figure 8* and *Figure 8—figure supplements 2* and *3* applied to *H. sapiens* and their corresponding Gene Ontology identifiers descriptions extracted from the Ensembl genome database.

- Transparent reporting form

### Data availability

All data generated during this study are included in the manuscript and supporting files. Source file for the code is available at https://github.com/BaderLab/Cell_Cycle_Theory (copy archived at https://archive.softwareheritage.org/swh:1:rev:7eb38b679e917ba8522b17edae5498990a221ffc).

The following previously published datasets were used:

| Author(s) | Year | Dataset title | Dataset URL | Database and Identifier |
|---|---|---|---|---|
| Yuzwa SA, Borrett MJ, Innes BT, Voronova A, Ketela T, Kaplan DR, Bader GD, Miller FD | 2017 | Developmental emergence of adult neural stem cells as revealed by single cell transcriptional profiling | https://www.ncbi.nlm.nih.gov/geo/query/acc.cgi/GSE107122 | NCBI Gene Expression Omnibus, GSE107122 |

| Briggs JA, Weinreb C, Wagner DE, Megason S, Peshkin L, Kirschner MW, Klein AM | 2018 | The dynamics of gene expression in vertebrate embryogenesis at single cell resolution | https://www.ncbi.nlm.nih.gov/geo/query/acc.cgi/GSE113074 | NCBI Gene Expression Omnibus, GSE113074 |
|---|---|---|---|---|
| Wagner DE, Weinreb C, Collins ZM, Megason SG, Klein AM | 2018 | ystematic mapping of cell state trajectories, cell lineage, and perturbations in the zebrafish embryo using single cell transcriptomics | https://www.ncbi.nlm.nih.gov/geo/query/acc.cgi/GSE112294 | NCBI Gene Expression Omnibus, GSE112294 |
| Geirsdottir L, David E, Keren-Shaul H, Weiner A, Bohlen SC, Neuber J, Balic A, Giladi A, Sheban F, Dutertre C-A, Pfeifle C, Peri F, Raffo-Romero A, Vizioli J, Matiasek K, Scheiwe C, Meckel S, Mätz-Rensing K, Thormodsson FR, Stadelmann C, Zilkha N, Kimchi T, Ginhoux F, Ulitsky I, Erny D, Amit I, Prinz M | 2019 | Cross-species analysis across 450 million years of evolution reveals conservation and divergence of the microglia program | https://www.ncbi.nlm.nih.gov/geo/query/acc.cgi/GSE134707 | NCBI Gene Expression Omnibus, GSE134707 |

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

## Appendix 1

### Why is cell cycle duration changing?

While defining a general mathematical representation of cell cycle kinetics for a developing system, we assembled available cell cycle length measurements from published studies for various species and tissues. *Figure 1* shows measurements obtained from *M. musculus*. For other data, see *Supplementary file 1*. The data motivated us to ask 'why is cell cycle duration changing over development?' and propose that changes in cell cycle duration can be used to guide the progression of cell development.

Theoretically we devised a simple model that can test this idea by assuming

- Cell cycle duration can change across developmental time
- Gene length distribution is constant among all cells in the same organism, such that we can denote the length by L
- The difference in cell cycle can affect the time a cell spends transcribing genes
- All active genes are transcribed and transcription rate is constant in a cell

The novel aspect of our work is the proposal that a cell cycle-dependent transcriptional filter can control cellular diversity within a tissue over development. However, some of the concepts that we build on are known and are recognized in the community to varying degrees. We bring these together for the first time to support the model and generate predictions. In particular, we list these concepts below and clarify our novel contribution.

Prior contributions:

- Cell cycle lengthens over development.
  - 'The *Xenopus embryo* undergoes 12 rapid synchronous cleavages followed by a period of slower asynchronous divisions more typical of somatic cells after which the cell cycle duration continues to increase.' (*Newport and Kirschner, 1982a*); https://pubmed.ncbi.nlm.nih.gov/6183003
  - 'In *D. melanogaster* early development, the first 10 cell divisions are fast and synchronous, then progressively increase in cell cycle duration.' (*Foe, 1989*; *Foe and Alberts, 1983*); https://pubmed.ncbi.nlm.nih.gov/6411748; https://pubmed.ncbi.nlm.nih.gov/2516798
  - The cell cycle lengthens during *M. musculus* brain development. 'The length of the cell cycle increases from 8.1 to 18.4 hr, which corresponds to a sequence of 11 integer cell cycles over the course of neuronal cytogenesis in mice. The increase in the length of the cell cycle is due essentially to a fourfold increase in the length of G1 phase which is the only phase of the cell cycle which varies systematically.' (*Takahashi et al., 1995a*); https://pubmed.ncbi.nlm.nih.gov/7666188
  - We also compiled cell cycle duration from 25 papers, which clearly support this statement (see *Figure 1* and *Supplementary file 1*).
- Gene length controls transcription timing. Short cell cycles limit transcription and long cell cycles allow transcription of longer genes.
  - Cell cycle duration can limit transcripts based on their size
    - Short cell cycles can constrain transcription in *D. melanogaster*. 'The length of mitotic cycles provides a physiological barrier to transcript size, and is therefore a significant factor in controlling developmental gene activity during short "phenocritical" periods.' (*Rothe et al., 1992*); https://pubmed.ncbi.nlm.nih.gov/1522901
  - Zygotic transcripts are encoded by short genes and start being expressed when cell cycle lengthens.
    - 'We propose that early development in *Drosophila* operates according to a hierarchy of events. The first 13 division cycles are driven by a maternal mechanism which responds to the increasing nuclear density by extending the interphase periods of successive cycles. This lengthening of interphases allows transcriptional activation, and the expression of new zygotic gene products triggers events such as cellularization of the blastoderm, gastrulation, and further rounds of mitosis.' (*Edgar et al., 1986*); https://pubmed.ncbi.nlm.nih.gov/3080248
    - *D. rerio* zygotic transcript lengths are shorter than maternally provided ones; the earliest zygotic genes are without introns. (*Heyn et al., 2014*; *Kwasnieski et al., 2019*; *Shermoen and O'Farrell, 1991*); https://pubmed.ncbi.nlm.nih.gov/1680567; https://pubmed.ncbi.nlm.nih.gov/24440719; https://pubmed.ncbi.nlm.nih.gov/31235656

- - Longer genes, with larger introns, take longer to transcribe ('intron delay').
    - Intron delay and transcriptional timing can affect development. (*Artieri and Fraser, 2014*; *Gubb, 1986*; *Swinburne and Silver, 2008*); DOI:10.1002/dvg.1020070302; https://pubmed.ncbi.nlm.nih.gov/18331713; https://pubmed.ncbi.nlm.nih.gov/2506953
- Cell cycle-dependent transcriptional filter is a mechanism for gene transcript expression regulation.
  - Hypothesized in *Bryant and Gardiner, 2016*, but no analysis or experimental data to support this statement is provided in this publication.
- Cell cycle length changes can control cell fate and development.
  - In cell lines
    - Differentiation can be induced in G1-phase isolated pluripotent embryonal carcinoma cells by treating with retinoic acid (RA) while other cell cycle phases are not RA stimulated. (*Mummery et al., 1987*); https://pubmed.ncbi.nlm.nih.gov/2883052
    - 'A short G1 phase is an intrinsic determinant of naïve embryonic stem cell pluripotency.' (*Coronado et al., 2013*); https://pubmed.ncbi.nlm.nih.gov/23178806
    - 'The cell cycle state of stem cells determines cell fate propensity.' (*Pauklin and Vallier, 2013*); https://pubmed.ncbi.nlm.nih.gov/24074866
    - Embryonic stem cells are more responsive to differentiation signals in G1 than in other phases of the cell cycle. (*Singh et al., 2013*); https://pubmed.ncbi.nlm.nih.gov/24371808
  - In an organism
    - Alteration of cell cycle length can cause changes in *Gallus gallus* limb pattern. Gene transcripts normally expressed in mesenchyme cells are sensitive to cell cycle length. (*Ohsugi et al., 1997*) ; https://pubmed.ncbi.nlm.nih.gov/9281333
- Transcription rate and duration
  - Transcription elongation rate is about 1.4 kb/min
    - Transcript elongation rates tend to be uniform within a cell type. (*Ardehali and Lis, 2009*) ; https://pubmed.ncbi.nlm.nih.gov/19888309
    - Transcription of human dystrophin gene requires 16 hr. (*Tennyson et al., 1995*); https://pubmed.ncbi.nlm.nih.gov/7719
  - Transcription is repressed in S phase.
    - Early evidence that transcription is repressed in synthetic phase (S). (*Newport and Kirschner, 1982b*) ; https://pubmed.ncbi.nlm.nih.gov/7139712
    - 'Upon G1–S transcriptional activation, cells progress to S phase, initiate DNA replication, and subsequently inactivate transcription.' (*Bertoli et al., 2013*); https://pubmed.ncbi.nlm.nih.gov/23877564

## Our novel contributions

- Our main novel claim: we are the first to link cell cycle duration to control of cell diversity and proportions of cells in tissues.
- We are the first to support the idea that a cell cycle-dependent transcriptional filter is a mechanism for gene transcript expression regulation that affects development using quantitative modeling.
- We are the first to link gene length distribution in genomes of multiple species to length of organism development.
- We are the first to show major functional differences between short and long genes in animal genomes.
- Our single-cell transcriptomic mathematical model is novel and shared as a community resource.

