## [Decision Letter]

**Acceptance summary:**

Cell cycle duration acting as a filter that constrains transcription is an idea proposed 30 years ago. Here, authors propose that a very simple model can produce results that qualitatively echo single-cell RNA seq data published by other labs. Overall, this study suggests that the slowing down of the cell cycle during development can act to allow longer genes to be transcribed and more cell types to be generated. Experimental test of this hypothesis are needed for future work.

**Decision letter after peer review:**

[Editors’ note: the authors submitted for reconsideration following the decision after peer review. What follows is the decision letter after the first round of review.]

Thank you for submitting your work entitled "Control of tissue development by cell cycle dependent transcriptional filtering" for consideration by *eLife*. Your article has been reviewed by 4 peer reviewers, including Wenying Shou as the Reviewing Editor and Reviewer #1, and the evaluation has been overseen by a Senior Editor. The following individual involved in review of your submission has agreed to reveal their identity: David M Suter (Reviewer #2).

Our decision has been reached after consultation between the reviewers. Based on these discussions and the individual reviews below, we regret to inform you that your work will not be considered further for publication in *eLife*.

Although there was some support for your work, the amount of time required for revision will likely exceed two months. It is the policy of *eLife* that a manuscript requiring substantial revision should be rejected. We do invite you to resubmit if you feel that you have addressed reviewers comments.

*Reviewer #1:*

Abou Chakra et al. posed the hypothesis that cell cycle transcriptional filtering – the transcribing of long genes only when cell cycle slows down – might help control tissue development and the generation of diverse cell fates. This study is based on simple math modeling and comparing model predictions to single-cell RNA seq data.

From an outsider's point of view, the article is interesting, although very speculative. Thus, I suggest softening your statements throughout. For example, the current title can be changed to "Cell cycle dependent transcription filtering can potentially contribute to tissue development".

Scientifically, I would like to see an analysis of whether transcripts specific to development (e.g. neuronal development) tend to be longer.

*Reviewer #2:*

This study addresses the important and exciting topic of how the cell cycle duration gates gene expression diversity at the single cell level by limiting the expression of long genes. This work intersects between genetics, gene expression, developmental and evolutionary biology, and should therefore be of broad interest to the readership of *eLife*. The manuscript is well written and structured, and the findings are interesting. I have some technical and conceptual concerns about the causality links made by the authors.

1. In general the authors should provide more details about the implementation of their method in the manuscript. While Figure 1 is very clear and allows non-specialists to grasp the concept, a more detailed explanation of the model would be useful.

In particular, how is transcription re-initiation modelled ? This is important for the following reason. If a gene takes 10 hours to be transcribed and the cell cycle is 10 hours, only 1 transcript can be made. However, if the cell cycle is longer, the maximum number of transcripts will depend on the distance between successive polymerases (thus depending on initiation rate), and could thus rapidly increase. The number of transcripts generated in 11 hours will thus scale with initiation rate, or inversely with the distance between polymerases on the gene. Also, how does transcriptional bursting affect their modelling ?

2. The authors show that short genes tend to be expressed at higher levels than long genes. They use this data to support that elongation rates * cell cycle duration limits the expression of long genes. In this scenario, one would expect that long genes are more depleted in pol II at the end of the gene, while short genes would not. The authors could look into Pol II footprinting/ChIP-seq datasets to confirm this, and also to exclude that short genes are expressed at higher levels because of their higher initiation rates.

3. The authors mention that cell cycle duration should be broadly correlated with development time. But cell sizes and organism size will also impact this correlation. Why do the authors not consider these parameters ?

4. The authors should provide an analysis of which classes of genes are enriched into different genes length bins. According to their results, more specialised genes, i.e. expressed in terminally differentiated, non-dividing cells should be longer on average.

5. Figure 6A: How can the average cell cycle duration become higher than the max cell cycle duration at the single cell level (shown as color code) ?

6. What is the relative contribution of introns vs exons in long vs short genes ? Could the longer cell cycle of some species allow to accumulate more/longer introns to increase splice isoform diversity/regulatory potential ?

7. Linked to 5., one challenge here is to understand causality links. The cell cycle could be used by organisms to gate cell diversity during development, but longer cell cycles could also allow to accumulate longer genes on evolutionary time scales. The authors should comment on this.

*Reviewer #3:*

We've known for a very long time that cells in many animal embryos have very fast cell cycles, and that the duration of cell cycle increases as cells differentiate into progeny with specialized function. However, whether cell cycle duration directly impacts cell fate decisions via filtering transcriptional activity is not clear. Certainly it has been proposed on multiple occasions, and the ability of mitosis and DNA replication to interrupt and abort transcription has been demonstrated. This study presents an interesting attempt to address this topic via mathematically modeling the relationship of cell cycle length and the diversity of transcriptome. The authors utilized a simplified model to simulate how cell cycle length, gene length, number of genes and transcription rate affect resulting transcriptomes in cell populations. Not unexpectedly, the simulations show that increasing the length of the cell cycle can increase the proportion of mRNAs from long genes, and also the complexity of the transcriptome and (more interestingly) the diversity of transcriptomes between different cells in a population. The model is clever and the demonstration is useful, if highly over-simplified. Importantly the authors also analyze some real transcriptome data to see if it supports their conclusions. At some levels there is support, but overall the real data from single cell sequence don't align well with their hypothesis and fall short of a compelling experimental validation. Overall we felt the study was interesting in concept and could be a valuable addition to the literature devoted to cell cycle and development, but that it could be much better with comprehensive revisions as discussed below.

1. The simulations in Figure 2A show that increasing cell cycle duration will allow more relative transcription of longer genes. Unfortunately the real data from *Xenopus* and Danio (Figure 2B, S2) don't appear to show this same trend. The authors should check this more carefully by comparing proportions of transcripts of different lengths at the different developmental times. If there is not change in proportions then the real data may not validate the simulation's predictions. At issue might be the "contamination" of the real data with maternal transcripts. Perhaps the authors should try to remove maternal transcripts and analyze only zygotic transcripts. Furthermore, the timepoints shown for *Xenopus* in Figure 2B are probably too late and too closely spaced to see a trend. They should compare transcripts for very early and very late in development.

2. I was not convinced by the simulations and arguments about the generation of cell diversity. It seems that this might only occur with very low numbers of transcripts per cell; i.e. when stochastic variation came into effect. This may not be the case for real cells.

3. Figure 6AB really baffled me. Transcriptome diversity seems not to be graphed at all (check the axis labels and key), even though diversity is the point of the figure. Either it is mis-labeled or it needs more explanation. Figure 6C also requires more explanation, both within the figure and in the text (lines 219-227), which is opaque.

4. Figure 5 states that slower cell cycles would increase cell type diversity at the price of fewer progeny number. However, the real data in Figure 6 don't support this idea. Slower cell cycle actually increased differentiated progenies, this suggests that the authors' simulation settings failed to capture a critical aspect of the regulation of transcription.

5. We suggest that the authors also look into the Oikopleura dioica genome (see Danks et al. 2012 "OikoBase: a genomics and developmental transcriptomics resource for the urochordate Oikopleura dioica"). This could be very interesting because this organism develops exceptionally fast (embryogenesis last 4-5 hours before hatching a tadpole), and they have an incredibly compact genome (most introns are no longer than 100 bp). There is also good transcriptome data from *Drosophila* that could be informative, especially if maternal transcripts could somehow be subtracted out. Overall, the analysis of real transcriptome data is superficial, and much more could be done here, that might support the authors' conclusions much better that what is presented.

6. In the introduction (line 47), the authors should state how they envision the "transcriptional filter" works. By abortion of transcription at M phase? Or during S phase? There is data on both mechanisms and these should be cited and described explicitly. This also comes up in the discussion (line 269). The authors should be aware that the attenuation of transcription during S-phase is limited to interactions with replication forks. In fact there is a great deal of transcription in most S phases.

7. We were uncomfortable with the assumption that appears to be made on lines 86-92 and Figure 2, where the number of transcripts made during a period of time equals: synthesized transcripts=time/time to make one transcript. Does this assumption fail to take into account that multiple RNA transcription bubbles may exist on the same gene once transcription starts, thus speeding up transcript production once the first transcript in finished? If so it is inaccurate.

*Reviewer #4:*

This manuscript presents a theoretical model that explains potential contribution of cell cycle, as a transcription filter, to organism development. The hypothesis is not entirely new. The main contribution is that the authors defined a math/simulation framework and compared it with some real data (gene size distribution in various organisms and single-cell transcriptomic data obtained from several organisms/processes) for justification. Overall, I do not feel this work adds much new to current understanding. Real data did not strongly validate the proposed model, nor led to refinement in knowledge/hypotheses.

1. Page 4. Algorithms and parameters used in simulation were not described in sufficient detail. For example, how were single-cell RNA-seq data simulated? Was noise considered?

2. Figure 2, the trends between A and B are similar but the actual distributions are not: either mean average transcript count per cell, or the extent of spread look quite different. One could also argue the dissimilarity between the two figures. Does it reveal that the model lacks necessary accuracy? This needs to be further discussed/explained.

3. Figure 3A. The authors should be able to derive analytical solutions for the transcriptome diversity, directly from the model defined in Figure A. Not sure why they need to show these relations indirectly from simulation (Page 5). Perhaps it is the order/logic of the presentation.

4. Figure 3C. Unclear how many cells are there under each condition (when does the simulation stop?) and how do #cell affect the clustering results. Also, #cluster, as a metric is quite misleading (not comparable across conditions) without specifying the total #cell and the clustering algorithms/parameters used.

5. Page 6, the authors stated that "we expect faster developing organisms to have short cell cycles and genes whereas slower developing organisms will have longer cell cycles and genes". This is very rough. Can it be quantified and then supported by the proposed model?

6. Figure 4. How were the 11 genomes selected? Why not select more? Are there genomes that do not follow the trend, i.e., having large genome but relatively shorter genes? Also, it is unclear in what order the organisms are listed. Particularly, Fugu and zebra-fish did not follow the monotonic trend in means. Some distributions do not look statistically significantly different.

7. Figure 6. E15.5 and E17.5 has a slightly reversed trend.

In general, there needs to be more real data used to back up the theoretical models.

[Editors’ note: further revisions were suggested prior to acceptance, as described below.]

Thank you for submitting your article "Control of tissue development and cell diversity by cell cycle dependent transcriptional filtering" for consideration by *eLife*. Your article has been reviewed by 3 peer reviewers, including Wenying Shou as the Reviewing Editor and Reviewer #1, and the evaluation has been overseen by Aleksandra Walczak as the Senior Editor. The following individual involved in review of your submission has agreed to reveal their identity: David M Suter (Reviewer #2).

The reviewers have discussed their reviews with one another, and the Reviewing Editor has drafted this decision letter to help you prepare a revised submission.

Essential revisions:

The decision has taken a long time because as you will see, the three reviewers have disagreements. Upon further discussions, we have reached an agreement. We feel that although experiments are always desired, there should be a place in science for extracting information from published datasets, despite the varying quality of datasets. We will waive the requirements for experiments, but we do request you to address comments not related to experiments, and be very careful in stressing the limitations of the datasets you used and the conclusions you drew. For example, your model suggests a mechanism, but does not exclude other mechanisms.

*Reviewer #1:*

1. Figure 4: It may be more meaningful to model stem-cell like behavior where a fast cell always gives birth to a slow and a fast cell, whereas a slow cell always gives rise to two slow cells. I believe that the cell # patterns will look more realistic under this assumption.

2. Figure 7: I am not sold about this figure and the associated text. "Sensory" and "perception" (short genes) seem to be related to neural-development (long genes). Also, the main text said "shortest genes… enriched for genes involved in core processes (e.g…. transcription…), whereas in Figure 7, "transcription" is associated with long genes.

*Reviewer #2:*

I found the revised version of this manuscript improved, and the authors have adequately addressed most of the points I raised.

Notably, they now describe more in detail their methodology, and also assess how their model behaves when assuming that several RNA Pol II can be present on the same gene. I would suggest to use this as a first assumption rather than "We assume RNA polymerase II re-initiation occurs once a transcript is complete". To me the latter one is not supported by what we know about transcription that occurs in bursts that can generate large number of RNA molecules within minutes. Also see Tantale et al., Nature Communications 2016, who show evidence of RNA Pol II convoys on actively transcribed genes.

*Reviewer #3:*

In this manuscript the authors use mathematical modeling to address whether cell cycle length determines cell fate using a correlation of gene transcript length. Since a longer cell cycle time, allows transcription of longer genes, it could affect the cell fate of the progeny. If longer transcripts are needed for highly differentiated cells, there would be a need for longer cell cycle times. Since it has been shown in stem cells that lengthening of the G1 phase is correlated with increased differentiation of cells, this hypothesis could make a lot of sense.

Using mathematical modeling is a great approach to answer this question and is definitely one of the strengths of this manuscript. This manuscript is trying to address an important and fundamental question that has been on the minds of scientists for a long time.

The drawback of the manuscript is that validation of the hypothesis is only partially or poorly confirmed by the experimental data. Essentially, the data does not contradict the hypothesis of the authors. This is great but is it good enough? Should the data not univocally prove that the hypothesis is correct? One of the major issues is that the authors use publicly available data, which originates from different organisms, different developmental time points, and have been acquired using different platforms. Therefore, the underlying data may not be solid enough.

Rather than trying to find universal rules that apply to all organisms, tissues, and developmental time points, it may be more useful to stick to one organism. If the authors could prove that their hypothesis is correct even in only one specific cell types, this would be an important step. Sometimes taking a small step can be more important than making a giant leap that is not well supported by the data.

This manuscript is interesting and contains good hypotheses but for sure the authors had to use a number of simplifications. Whether this still allows to generalize the conclusions of this manuscript is up for debate.

I am not a mathematician and therefore I am not able to check the mathematical models that were used. Nevertheless, I will assess if the conclusions make sense in real biology.

My conclusion after reading this manuscript is that of interest but remains speculative. What I mean by this is that the mathematical predictions would need to be verified by experiments. Although the authors use a number of datasets, they are assembled from different organisms and different developmental timepoints. As the authors mention, the data does not contradict their hypotheses. This is ok but maybe not good enough? Should the data not univocally support the mathematical hypotheses in order that the readers will buy them?

Here are the main reasons:

1. Line 128: "in general, cells express more short genes than longer genes over multiple developmental time points." Although there may be a trend, I am not entirely convinced of this statement. There seems to be a lot of noise (variation), which may not support this conclusion.

2. Line 223: "While cell cycle duration measurements are not widely available, we instead ask if organisms with longer genes would also take longer to develop." Although this is understandable, I am not sure that this is a correct surrogate. The duration of development must not necessarily be dependent on cell cycle length. Nevertheless, I agree the cell cycle duration measurements are not widely available.

3. The authors use data from different organisms and from different developmental time points. Of course, the idea is that there are universal rules that apply across species. This would be ideal but is there any proof of that? The unwanted side effect is that it becomes really confusing and the authors may compare apples to oranges.

4. Then there is the issue of splicing and introns. It is not surprising that larger genes contain more introns. To some degree splice isoforms could also explain the differences between stem cells and differentiated cells. Nevertheless, I feel this is a distraction. Therefore, analyzing organisms that contain few introns would be more useful. Budding yeast is such an organism.

5. The pathway analysis of the short and long genes is not thorough enough. In addition, the authors should use random sets of genes (same number) from the intermediate genes, which are the majority of genes.

6. The time it takes to transcribe a gene is not only dependent on its size and a fixed speed. This is an oversimplification.

[Editors’ note: further revisions were suggested prior to acceptance, as described below.]

Thank you for submitting your article "Control of tissue development and cell diversity by cell cycle dependent transcriptional filtering" for consideration by *eLife*. Your article has been reviewed by 3 peer reviewers, including Wenying Shou as the Reviewing Editor and Reviewer #1, and the evaluation has been overseen by Aleksandra Walczak as the Senior Editor. The following individual involved in review of your submission has agreed to reveal their identity: David M Suter (Reviewer #2).

Essential revisions:

Please revise your writing to address Reviewer 1 and 3's critiques.

*Reviewer #1:*

Authors have mainly addressed my comments.

Figure 9: I wonder whether you can make further statements. For example, if immune cells have short cell cycle, then its enrichment for short genes will make more sense. Also, might olfactory short genes be related to environmental sensing genes which in turn involve signal transduction pathways also used in fast-growing cells?

Figure 5 legend: 2^20^ should be 2^19^.

*Reviewer #2:*

I am happy to see that the authors successfully integrated RNA Pol II re-initiation in their model, which did not affect their conclusions. The authors have addressed my concerns adequately, and the manuscript should be ready for publication.

*Reviewer #3:*

The authors have invested efforts to address the issues that were raised by the reviewers. The story of this manuscript has not fundamentally changed (which probably was also not expected) and there remain shortcomings. One aspect that I wish would improve is to use more understatement rather than claiming things that the authors cannot prove.

Here are a few examples, there the manuscript could be improved:

1. Line 128/129: "found that, in general, short genes have a higher expression level than longer genes within a cell." When I was reading this, I had trouble believing it but in Figure 3B, the authors show mRNA expression. This is though not mentioned in the text and the reader can be misled that this also applies to protein expression. It would be desirable that the authors are precise without using generalizations.

2. Line 179: "second child cell" I believe these are usually referred to as "daughter cells".

3. Line 233: "We started by asking if organisms with longer genes would also take longer to develop." I apologize but this question (or hypothesis) does not make a lot of sense to me. There are a million reasons why an organism takes a certain amount of time to develop and this may be also dependent on the environment. Reducing it to the length of the genes is surely only one of many reasons. In their conclusion on line 249, the authors call it "strong relationship", which probably is an association and we all know that associations are weak (remember the one about the amount of chocolate consumption and that chance to win the Nobel prize?).

4. Line 266-278: I am not sure if I get the point here "cell cycle duration and gene expression vary spatially.". Not only spatially but also dependent on age, environment, nutrition, and many more factors.

5. In the discussion, the limitations (some of which are mentioned) should be discussed much more honestly.

6. In several figures (for example Figure 6—figure supplement 2 but there are others), the authors use a representation (word clouds) that are not very helpful. The authors should find a better way to bring across the point that they are trying to make.

---

## [Author Response]

[Editors’ note: the authors resubmitted a revised version of the paper for consideration. What follows is the authors’ response to the first round of review.]

Reviewer #1:Abou Chakra et al. posed the hypothesis that cell cycle transcriptional filtering – the transcribing of long genes only when cell cycle slows down – might help control tissue development and the generation of diverse cell fates. This study is based on simple math modeling and comparing model predictions to single-cell RNA seq data.From an outsider's point of view, the article is interesting, although very speculative. Thus, I suggest softening your statements throughout. For example, the current title can be changed to "Cell cycle dependent transcription filtering can potentially contribute to tissue development".Scientifically, I would like to see an analysis of whether transcripts specific to development (e.g. neuronal development) tend to be longer.

Thank you for the comments. The major change we made to address this is to focus the paper on the mathematical model and separate the speculative statements into a new section that includes discussion of the implications of the model, providing a number of new analyses to support these implications. We have also reviewed the paper as a whole to clarify speculative statements. We hope this clarifies and strengthens the overall paper.

We have also included a pathway enrichment analysis based on gene length which clearly shows that the longest genes are enriched for genes expressed in mature cells (long cell cycles) in processes such as neural and muscle development, whereas the shortest genes are enriched for genes involved in core processes (e.g. metabolism, transcription, translation) and transcription factors.

Reviewer #2:This study addresses the important and exciting topic of how the cell cycle duration gates gene expression diversity at the single cell level by limiting the expression of long genes. This work intersects between genetics, gene expression, developmental and evolutionary biology, and should therefore be of broad interest to the readership of eLife. The manuscript is well written and structured, and the findings are interesting. I have some technical and conceptual concerns about the causality links made by the authors.1. In general the authors should provide more details about the implementation of their method in the manuscript. While Figure 1 is very clear and allows non-specialists to grasp the concept, a more detailed explanation of the model would be useful.

Thank you we have now included a more detailed explanation in the methods section.

In particular, how is transcription re-initiation modelled ? This is important for the following reason. If a gene takes 10 hours to be transcribed and the cell cycle is 10 hours, only 1 transcript can be made. However, if the cell cycle is longer, the maximum number of transcripts will depend on the distance between successive polymerases (thus depending on initiation rate), and could thus rapidly increase. The number of transcripts generated in 11 hours will thus scale with initiation rate, or inversely with the distance between polymerases on the gene. Also, how does transcriptional bursting affect their modelling ?

In our original manuscript, we assumed that a transcript must be fully transcribed before another copy can be made. That is, we didn’t allow a transcript to be transcribed by multiple polymerases at the same time. Including an initiation rate would act to increase the number of transcripts as the reviewer points out. To show how this affects our results, we performed a simulation with re-initiation, re-initiation distance is the kb distance an rnaPol complex can reinitiate along the gene, (Figure S2A, e.g. if it is set to 1, a new rnaPolII can initiate 1 kb apart from the first) or a random transcription rate (Figure S2B, random rate means a new rate is assigned to each gene from a Gaussian distribution around a set rate, λ). We show that changing initiation distance and transcription rate doesn’t affect the trend that short genes are expressed more than long genes (Figure S2 A-B). Furthermore, we test under which condition the transcriptional filter can be suppressed. We conducted simulations for a genome with genes of the same length (e.g. L_G_=9) and the cell cycle duration is short Γ=1 hour (Figure S2 C-D). In this case the long genes do not have enough time to be transcribed and changing initiation distance cannot overcome the limiting effects of the cell cycle. Only by increasing the transcription rate sufficiently high (e.g. λ=4) do we see the genes completely transcribed. On the other hand, simulation for a genome with genes of the same length (e.g. L_G_=9) with a long cell cycle duration (Γ=10 hours) (Figure S2 E-F ) shows that re-initiation distance and transcriptional rate help make cell cycle duration more important.

2. The authors show that short genes tend to be expressed at higher levels than long genes. They use this data to support that elongation rates * cell cycle duration limits the expression of long genes. In this scenario, one would expect that long genes are more depleted in pol II at the end of the gene, while short genes would not. The authors could look into Pol II footprinting/ChIP-seq datasets to confirm this, and also to exclude that short genes are expressed at higher levels because of their higher initiation rates.

We tried to do this by analyzing a series of Pol II footprinting data sets (we tried GSE34301 (Gaertner et al., 2012), GSE81521 (Li et al., 2016), and GSM565202 (Saha et al., 2011)), but the data did not distinguish between an elongating versus a poised RNA pol II. There were very few data sets of this type available for organism development and we couldn’t find any for early development to better match the context of our study. Also, we couldn’t be sure what phase of the cell cycle the cells were in when measured. Overall, the reviewer makes an interesting point and there do seem to be a number of technologies that may be able to support this line of inquiry (e.g. GRO-seq and related methods), so we will be watching for these data in the future.

3. The authors mention that cell cycle duration should be broadly correlated with development time. But cell sizes and organism size will also impact this correlation. Why do the authors not consider these parameters ?

We agree that cell size will impact the decision of when the cell will divide, however the data on how cell size is influenced by cell cycle duration is still unclear, and current work suggests that S-phase is important in cell size decision (Kafri et al., 2013). We had added this to the discussion.

4. The authors should provide an analysis of which classes of genes are enriched into different genes length bins. According to their results, more specialised genes, i.e. expressed in terminally differentiated, non-dividing cells should be longer on average.

We have now included a pathway enrichment analysis based on gene length which clearly shows that the longest genes are enriched for genes expressed in mature cells (that we expect to have long cell cycles) in processes such as neural and muscle development, whereas the shortest genes are enriched for genes involved in core processes (e.g. metabolism, transcription, translation) and transcription factors, see figures S7, S8 and S9.

5. Figure 6A: How can the average cell cycle duration become higher than the max cell cycle duration at the single cell level (shown as color code) ?

Sorry for the confusion. This may have been caused by our lack of a legend in Figure 6A, which is now included.

6. What is the relative contribution of introns vs exons in long vs short genes ?

To answer this question we calculated the ratio of intron length to total gene length for each human gene, and plotted the distribution for four gene length bins (all bins have roughly the same number of genes). This shows that short genes (n=1453, dashed line at 0.05 below) have relatively short or no intron regions, while the longest genes (dashed line at 0.95) have the greatest intron length-to-gene length ratio.

Could the longer cell cycle of some species allow to accumulate more/longer introns to increase splice isoform diversity/regulatory potential ?

This is an interesting question. We do expect this to happen. Other questions arise from this line of inquiry. For instance, how does gene length and intron/exon structure change over evolution? How do these changes affect different gene function categories? How do these relate, considering various factors including intron length, position, number of introns? We are working on these questions for a separate paper focused on these questions, so did not include these results here.

7. Linked to 5., one challenge here is to understand causality links. The cell cycle could be used by organisms to gate cell diversity during development, but longer cell cycles could also allow to accumulate longer genes on evolutionary time scales. The authors should comment on this.

Yes, this is an interesting challenge. We have generally assumed that gene lengths and cell cycle duration dynamics coevolved across evolutionary time scales; our hypothesis is that a cell cycle dependent transcriptional filter is a fundamental regulatory mechanism operating during development (because gene length is fixed in the genome). We have now included a comment about this in the discussion.

Reviewer #3:We've known for a very long time that cells in many animal embryos have very fast cell cycles, and that the duration of cell cycle increases as cells differentiate into progeny with specialized function. However, whether cell cycle duration directly impacts cell fate decisions via filtering transcriptional activity is not clear. Certainly it has been proposed on multiple occasions, and the ability of mitosis and DNA replication to interrupt and abort transcription has been demonstrated. This study presents an interesting attempt to address this topic via mathematically modeling the relationship of cell cycle length and the diversity of transcriptome. The authors utilized a simplified model to simulate how cell cycle length, gene length, number of genes and transcription rate affect resulting transcriptomes in cell populations. Not unexpectedly, the simulations show that increasing the length of the cell cycle can increase the proportion of mRNAs from long genes, and also the complexity of the transcriptome and (more interestingly) the diversity of transcriptomes between different cells in a population. The model is clever and the demonstration is useful, if highly over-simplified.Importantly the authors also analyze some real transcriptome data to see if it supports their conclusions. At some levels there is support, but overall the real data from single cell sequence don't align well with their hypothesis and fall short of a compelling experimental validation.Overall we felt the study was interesting in concept and could be a valuable addition to the literature devoted to cell cycle and development, but that it could be much better with comprehensive revisions as discussed below.1. The simulations in Figure 2A show that increasing cell cycle duration will allow more relative transcription of longer genes. Unfortunately the real data from *Xenopus* and Danio (Figure 2B, S2) don't appear to show this same trend. The authors should check this more carefully by comparing proportions of transcripts of different lengths at the different developmental times. If there is not change in proportions then the real data may not validate the simulation's predictions. At issue might be the "contamination" of the real data with maternal transcripts. Perhaps the authors should try to remove maternal transcripts and analyze only zygotic transcripts. Furthermore, the timepoints shown for *Xenopus* in Figure 2B are probably too late and too closely spaced to see a trend. They should compare transcripts for very early and very late in development.

The reviewer makes a number of good points, but unfortunately, it seems we confused the presentation by including multiple developmental time points. We intended for this figure to just show that short genes are more expressed than long genes – irrespective of developmental time point or difference in cell cycle duration. Since we don’t know parameters such as mRNA degradation rate and how these change over time, we didn’t feel comfortable making more precise statements, such as comparing the ratio of short/long gene expression values between time points. We have simplified this figure using new data that shows one time point per species for a wider range of species. The overall message of Figure 2 is that short genes are more expressed than long genes. As the reviewer mentions, this is not unexpected, but is important to show to validate this aspect of our model.

2. I was not convinced by the simulations and arguments about the generation of cell diversity. It seems that this might only occur with very low numbers of transcripts per cell; i.e. when stochastic variation came into effect. This may not be the case for real cells.

We created additional simulations with more genes in the genome (10,100, 1000) per cell and show we get the same results. As shown in Figure S4.

3. Figure 6AB really baffled me. Transcriptome diversity seems not to be graphed at all (check the axis labels and key), even though diversity is the point of the figure. Either it is mis-labeled or it needs more explanation. Figure 6C also requires more explanation, both within the figure and in the text (lines 219-227), which is opaque.

Thank you for pointing this out and we apologize for the confusion. The point of this figure (now Figure 5) is not about transcriptome diversity. It is that when you allow the cell cycle duration to slow down over time and more naturally (randomly) vary, closer to what we understand happens in a real developing organism, the model generates a diverse set of cell proportions (represented by pie charts). So it is the diversity of the pie charts (the cell proportion patterns) generated that we want to highlight. This is interesting because it suggests cell cycle duration could help control cell type proportions during tissue development. We have completely redesigned this figure, which we hope shows this more intuitively. In particular, we show how the cell duration parameters/cell division events that vary over time generate in a diverse set of cells (cycling at different rates) shown in pie charts. We also compared these results to real mouse brain development data where we know how cell cycle changes (lengthens) over time, showing a similar trend.

The novel idea here is that just changing cell cycle dynamics can influence the proportion of fast to slow cycling cells (which we can think of as stem and differentiated cells, especially in the neural context). We have updated the text to clarify our idea.

4. Figure 5 states that slower cell cycles would increase cell type diversity at the price of fewer progeny number. However, the real data in Figure 6 don't support this idea. Slower cell cycle actually increased differentiated progenies, this suggests that the authors' simulation settings failed to capture a critical aspect of the regulation of transcription.

Sorry for the confusion between figures 5 and 6. Figure 5 (original manuscript) investigates simulations involving two cell lineages with two different, but constant cell cycle durations. This illustrates that asynchronous cell cycle duration can select a mix of cell diversity programs (not allowing cell cycle duration to increase over time).

Figure 6 (original manuscript) shows the results of a simulation where the cell cycle duration of all cells is increasing over time. This illustrates that changing cell cycle duration over time affects the proportion of cells generated with various cycling rates. We are able to compare the generation of cell proportion diversity to real mouse brain development data and show the same trend in simulated and real data, as discussed above.

We have redesigned these figures (now Figure 4 and 5) to more clearly show the different questions being asked.

We note that we did not compare the number of cell types in simulations and real data because we couldn’t guarantee a match in the range of cell durations present, and there could be artifacts with single cell RNA-seq data not accurately identifying all cell types and proportions, or sensitivity to scRNA-seq data clustering parameters. However, we do note that our simulations show that cell type diversity increases and then decreases as cell duration increases (see Figure 3B) and interestingly, we do see the same trend of gradual increase, then decrease in the real mouse brain development data (scRNA-seq) across the four time points (8 cell types at E11.5, 11 at E13.5, 16 at E15.5 and 9 at E 17.5).

5. We suggest that the authors also look into the Oikopleura dioica genome (see Danks et al. 2012 "OikoBase: a genomics and developmental transcriptomics resource for the urochordate Oikopleura dioica"). This could be very interesting because this organism develops exceptionally fast (embryogenesis last 4-5 hours before hatching a tadpole), and they have an incredibly compact genome (most introns are no longer than 100 bp).

We have added analysis of O. dioica genome to our analysis. Interestingly, its gene length distribution is shifted globally to be shorter, in agreement with a fast development process, refer to Figure 6. Thank you for this suggestion.

There is also good transcriptome data from Drosophila that could be informative, especially if maternal transcripts could somehow be subtracted out.

Thank you. We were able to find a recent multi-organism (n=10) scRNA-seq data set to add to our analysis to address this comment and the related comments from reviewer 4.

Overall, the analysis of real transcriptome data is superficial, and much more could be done here, that might support the authors' conclusions much better that what is presented.

The main challenges with published transcriptome data are the potential confounding technical factors (e.g. cell type bias from single cell dissociation workflows), low sensitivity and lack of coverage of many biological contexts. We compared to real data where we felt confident doing so and resisted in other areas. We now include discussion of the opportunity for more to be done here in the future. For now, we want to get the overall idea out there to help us connect with experimentalists who could help generate new data designed to answer specific questions raised by the model.

6. In the introduction (line 47), the authors should state how they envision the "transcriptional filter" works. By abortion of transcription at M phase? Or during S phase? There is data on both mechanisms and these should be cited and described explicitly. This also comes up in the discussion (line 269).

Thank you for raising this point. We have added these citations to address this.

*– Danio rerio* zygotic transcript lengths are shorter than maternally provided ones; The earliest zygotic genes are without introns. (Heyn et al., 2014; Kwasnieski et al., 2019; Shermoen and O’Farrell, 1991) https://pubmed.ncbi.nlm.nih.gov/1680567https://pubmed.ncbi.nlm.nih.gov/24440719 https://pubmed.ncbi.nlm.nih.gov/31235656

– Cell cycle duration can limit transcripts based on their size

Short cell cycles can constrain transcription in *D. melanogaster*. “the length of mitotic cycles provides a physiological barrier to transcript size, and is therefore a significant factor in controlling developmental gene activity during short 'phenocritical' periods.” (Rothe et al., 1992) https://pubmed.ncbi.nlm.nih.gov/1522901

The authors should be aware that the attenuation of transcription during S-phase is limited to interactions with replication forks. In fact there is a great deal of transcription in most S phases.

We have not been able to find the extent of transcription in S phase, we have only found papers discussing that transcription was found to decrease at the onset of S phase (Newport and Kirschner, 1982b; Bertoli et al., 2013).

https://pubmed.ncbi.nlm.nih.gov/7139712

https://pubmed.ncbi.nlm.nih.gov/23877564

7. We were uncomfortable with the assumption that appears to be made on lines 86-92 and Figure 2, where the number of transcripts made during a period of time equals: synthesized transcripts=time/time to make one transcript. Does this assumption fail to take into account that multiple RNA transcription bubbles may exist on the same gene once transcription starts, thus speeding up transcript production once the first transcript in finished? If so it is inaccurate.

This point (also raised by reviewer 1 and discussed above) is now addressed by adding simulations that consider multiple RNA Pol II/transcription bubbles per transcript, see Figure S2. Adding this to our simulations does not change our conclusions.

Reviewer #4:This manuscript presents a theoretical model that explains potential contribution of cell cycle, as a transcription filter, to organism development. The hypothesis is not entirely new. The main contribution is that the authors defined a math/simulation framework and compared it with some real data (gene size distribution in various organisms and single-cell transcriptomic data obtained from several organisms/processes) for justification. Overall, I do not feel this work adds much new to current understanding. Real data did not strongly validate the proposed model, nor led to refinement in knowledge/hypotheses.

The novel aspect of our work is the proposal that a cell cycle dependent transcriptional filter can control cellular diversity within a tissue over development. However, some of the concepts that we build on are known and are recognized in the community to varying degrees. We bring these together for the first time to support the model and generate predictions. In particular, we list these concepts in the Appendix and clarify our novel contribution.

Our work is the first to tie these diverse ideas together into one unified model. Further, our model enabled us to uncover a novel emergent property, that a cell cycle controlled transcriptional filter can be used as a mechanism to control generation of diversity of cells within a developing system (one formed by successive rounds of cell division). Finally, we are the first to apply mathematical modeling to this topic, which enables us to propose testable predictions that we hope will interest experimental biologists to investigate.

Our comparison to real data cannot prove our model, and this is not used to justify, prove or validate our results. Instead, it serves to show that real data does not refute our predictions. In discussions with diverse experimental biologists, we have found comparison with real data in this way necessary to move forward with discussions about our predictions, and that is what we hope to accomplish by including it here.

We have edited the text to clarify the novelty and we have included the information in the Appendix.

1. Page 4. Algorithms and parameters used in simulation were not described in sufficient detail. For example, how were single-cell RNA-seq data simulated? Was noise considered?

Yes, noise is considered by the model in the cell division step, with random assortment of transcripts to children cells. We have clarified this and other aspects of the method in the manuscript, as also requested by reviewer 1.

2. Figure 2, the trends between A and B are similar but the actual distributions are not: either mean average transcript count per cell, or the extent of spread look quite different. One could also argue the dissimilarity between the two figures. Does it reveal that the model lacks necessary accuracy? This needs to be further discussed/explained.

The intent of figure 2 is only to show the overall pattern of higher expression in shorter genes than longer genes in both simulation and real data. We now make this clearer by showing one time point (in response to reviewer 3) and with data across species.

3. Figure 3A. The authors should be able to derive analytical solutions for the transcriptome diversity, directly from the model defined in Figure A. Not sure why they need to show these relations indirectly from simulation (Page 5). Perhaps it is the order/logic of the presentation.

The analytical solution for transcriptome diversity is described in the text. We include both simulation and calculations to show that they agree. We chose to focus on the simulation, because later in the paper, we consider number of cell clusters as an additional measure of transcriptome diversity and this is not straightforward to define an analytical solution for, so we choose to follow the simulation thread/explanation through the sections of the paper for consistency.

4. Figure 3C. Unclear how many cells are there under each condition (when does the simulation stop?) and how do #cell affect the clustering results.

There are 10,000 cells sampled from each simulation. Simulations were run over a single cell division event (stopping criteria), but were run 1,000,000 times to provide a good sampling of the space of solutions. All clustering results include exactly 10,000 cells, so they are comparable. We varied the number of cells (from 1000 up to 10,000) and obtained similar results. Including more cells reduced variance (we iterated n=20 for each the test) and as a result chose 10,000.

Author response image 2 shows how selecting (1000, 5000 or 10000) cells does not affect cluster number, shown for cell cycle durations 1-5 hours.

**Author response image 2. respfig2:** 

Also, #cluster, as a metric is quite misleading (not comparable across conditions) without specifying the total #cell and the clustering algorithms/parameters used.

Sorry for this oversight. We have now specified the number of cells (10,000 for each clustering result in the paper) and the clustering parameters (Using Seurat 3.1.2, cells were clustered using a shared nearest neighbor (SNN)-based Louvain algorithm with reduction set as ”pca”. The clustering resolution was set to 1 for all data sets, and all calculated PCs were used in the downstream clustering process using the original Louvain algorithm.)

5. Page 6, the authors stated that "we expect faster developing organisms to have short cell cycles and genes whereas slower developing organisms will have longer cell cycles and genes". This is very rough. Can it be quantified and then supported by the proposed model?

Thank you for highlighting this. We have now added discussion in a new text section and improved the figure (now Figure 6) to clarify this point.

6. Figure 4. How were the 11 genomes selected? Why not select more? Are there genomes that do not follow the trend, i.e., having large genome but relatively shorter genes? Also, it is unclear in what order the organisms are listed. Particularly, Fugu and zebra-fish did not follow the monotonic trend in means. Some distributions do not look statistically significantly different.

We originally selected these species to be distant to each other to cover a broad range of species, but also are common model organisms with well annotated genomes. The organisms are listed based on their evolutionary relationship and not the size of the genome. We have now analysed over 100 genomes (n=101) and we see the same trend: an increase in longer genes correlates with an increase in development duration (see Figure 6).

7. Figure 6. E15.5 and E17.5 has a slightly reversed trend.In general, there needs to be more real data used to back up the theoretical models.

We are only interested in interpreting the overall trend. This data is available as a time series and it is the only one where we also know the average cell cycle duration. We need both cell cycle and cell proportion information to compare data to simulation predictions. The discrepancy between E15.5 and E17.5 may be due to the difference in cell number, which is 5000 cells for E15.5 and 2000 cells for E17.5, however the prediction is that cell cycle duration can affect proportions and we see such an effect across the time series data.

[Editors’ note: what follows is the authors’ response to the second round of review.]

Essential revisions:The decision has taken a long time because as you will see, the three reviewers have disagreements. Upon further discussions, we have reached an agreement. We feel that although experiments are always desired, there should be a place in science for extracting information from published datasets, despite the varying quality of datasets. We will waive the requirements for experiments, but we do request you to address comments not related to experiments, and be very careful in stressing the limitations of the datasets you used and the conclusions you drew. For example, your model suggests a mechanism, but does not exclude other mechanisms.

Thank you for arranging another useful review of our work and for waiving the requirements for experiments. We have addressed all the other comments and concerns of the reviewers. A point-by-point response to the reviewers follows to support our resubmission.

Reviewer #1:1. Figure 4: It may be more meaningful to model stem-cell like behavior where a fast cell always gives birth to a slow and a fast cell, whereas a slow cell always gives rise to two slow cells. I believe that the cell # patterns will look more realistic under this assumption.

We have now added the “stem-cell like” behaviour of a fast cell giving rise to a slow and a fast cell as scenario 3 in Figure 4-now-5. These results are interesting. They show us how the organism can increase the number of “slow” cells in the system, at a cost of the number of fast cells in the system.

2. Figure 7: I am not sold about this figure and the associated text. "Sensory" and "perception" (short genes) seem to be related to neural-development (long genes). Also, the main text said "shortest genes… enriched for genes involved in core processes (e.g…. transcription…), whereas in Figure 7, "transcription" is associated with long genes.

We apologize for the confusion, you are correct that transcription occurs for both long and short genes, so we have removed this from the visualization and focused on showing terms that have an uneven distribution of gene lengths. We have extensively revised the pathway themes and reanalysed genes based on their length. For example, we separated olfactory, eye, auditory and brain related pathways. This more careful assignment of pathways to themes helps clean up our results. For example, we find ‘olfactory’, as opposed to the other 3 neural themes, has more (~500) short genes involved vs. long genes (~17).

We also differentiated pathways involved in brain development (structure/tissue), neurons, nerves, synapse/synaptic and neurotransmitters (e.g. dopamine, norepinephrine, serotonin). All these pathways still exhibit more long genes than short genes. The neuron specific pathways increase from ~15 short genes (<1.6kb) to ~400 long genes (>244kb). Although some (~50) short genes (<1600bp) are involved in brain related pathways, there are overall more (~500) long genes (>244kbp). Spine, neurotransmitters and synapse follow the same trend with more long genes than short genes.

We have now updated the Figure 7-now-8 and Figures 8-figures supplement 2-3 and the related text to incorporate these results. We have also added supplementary files 5 and 6 to show how each of the *H. sapiens* pathways were grouped in themes. Furthermore, we found that when *H. sapiens* genes were divided into 20 groups from shortest genes to the longest genes, the top themes associated with short genes (some examples shown in blue Author response image 3, with more in Figure 8—figure supplement 3) have a decreasing moving average across all gene groups, whereas top themes associated with longest genes (some examples shown in black in author response image 3, with more in Figure 8—figure supplement 3) have an increasing moving average across all gene groups.

**Author response image 3. respfig3:** Example gene function themes with strong trends associated with short (blue) or long (black) genes.

Reviewer #2:I found the revised version of this manuscript improved, and the authors have adequately addressed most of the points I raised.Notably, they now describe more in detail their methodology, and also assess how their model behaves when assuming that several RNA Pol II can be present on the same gene. I would suggest to use this as a first assumption rather than "We assume RNA polymerase II re-initiation occurs once a transcript is complete". To me the latter one is not supported by what we know about transcription that occurs in bursts that can generate large number of RNA molecules within minutes. Also see Tantale et al., Nature Communications 2016, who show evidence of RNA Pol II convoys on actively transcribed genes.

We now assume that several RNA Pol II can be present on the same gene by default. We repeated all simulations and generated new figures (3,4,5,6 3—figure supplement 1 and 6—figure supplement 1) with the assumption that several RNA Pol II can be present on the gene. This did not change our conclusions, but we’re happy that this is a more realistic model.

Reviewer #3:In this manuscript the authors use mathematical modeling to address whether cell cycle length determines cell fate using a correlation of gene transcript length. Since a longer cell cycle time, allows transcription of longer genes, it could affect the cell fate of the progeny. If longer transcripts are needed for highly differentiated cells, there would be a need for longer cell cycle times. Since it has been shown in stem cells that lengthening of the G1 phase is correlated with increased differentiation of cells, this hypothesis could make a lot of sense.Using mathematical modeling is a great approach to answer this question and is definitely one of the strengths of this manuscript. This manuscript is trying to address an important and fundamental question that has been on the minds of scientists for a long time.The drawback of the manuscript is that validation of the hypothesis is only partially or poorly confirmed by the experimental data. Essentially, the data does not contradict the hypothesis of the authors. This is great but is it good enough? Should the data not univocally prove that the hypothesis is correct? One of the major issues is that the authors use publicly available data, which originates from different organisms, different developmental time points, and have been acquired using different platforms. Therefore, the underlying data may not be solid enough.Rather than trying to find universal rules that apply to all organisms, tissues, and developmental time points, it may be more useful to stick to one organism. If the authors could prove that their hypothesis is correct even in only one specific cell types, this would be an important step. Sometimes taking a small step can be more important than making a giant leap that is not well supported by the data.This manuscript is interesting and contains good hypotheses but for sure the authors had to use a number of simplifications. Whether this still allows to generalize the conclusions of this manuscript is up for debate.I am not a mathematician and therefore I am not able to check the mathematical models that were used. Nevertheless, I will assess if the conclusions make sense in real biology.My conclusion after reading this manuscript is that of interest but remains speculative. What I mean by this is that the mathematical predictions would need to be verified by experiments. Although the authors use a number of datasets, they are assembled from different organisms and different developmental timepoints. As the authors mention, the data does not contradict their hypotheses. This is ok but maybe not good enough? Should the data not univocally support the mathematical hypotheses in order that the readers will buy them?

We have now added more explicit statements that address the limitation of the data used. We truly hope that publishing this work can help move us to be able to perform the experiments that are needed to test the model, by raising awareness of this idea.

Here are the main reasons:1. Line 128: "in general, cells express more short genes than longer genes over multiple developmental time points." Although there may be a trend, I am not entirely convinced of this statement. There seems to be a lot of noise (variation), which may not support this conclusion.

We have applied a statistical test and found that all of the comparisons of short gene expression to longer gene expression are highly significant (p-values <10-16, Kolmogorov-Smirnov test). This result has been added to the figure 3 caption.

2. Line 223: "While cell cycle duration measurements are not widely available, we instead ask if organisms with longer genes would also take longer to develop." Although this is understandable, I am not sure that this is a correct surrogate. The duration of development must not necessarily be dependent on cell cycle length. Nevertheless, I agree the cell cycle duration measurements are not widely available.

We are sorry for the confusion, we agree that duration of development does not necessarily imply cell cycle length. We changed the sentence accordingly.

3. The authors use data from different organisms and from different developmental time points. Of course, the idea is that there are universal rules that apply across species. This would be ideal but is there any proof of that? The unwanted side effect is that it becomes really confusing and the authors may compare apples to oranges.

It is correct that comparing patterns across species may be like comparing apples to oranges, though it does support the generality of the conclusion. On the other hand, comparing results within species is a more appropriate comparison, but may lack generality. We have thus included results for both complementary situations where possible. For instance, in Figure 3, we show results across species and 2—figure supplement 4 shows results within species for three different species. However, in general, we cannot say that all patterns are universal, so we have added phrasing in the discussion/future direction section about this.

4. Then there is the issue of splicing and introns. It is not surprising that larger genes contain more introns. To some degree splice isoforms could also explain the differences between stem cells and differentiated cells. Nevertheless, I feel this is a distraction. Therefore, analyzing organisms that contain few introns would be more useful. Budding yeast is such an organism.

We agree that splice isoforms can add to the differences seen among the cells. The model proposed is directed at total gene length, irrespective of whether introns are present or absent. However we reviewed over 380 genomes in the Ensembl database and identified 3 additional species with genomes that contain few introns (>85% of intronless genes in their genome) – these happen to all be fungi. We now include these (*Ashbya gossypii*, *Komagataella pastoris*, and *Yarrowia lipolytica*) in our analysis (figure 9). These genomes do show gene length dependent effects, similar to budding yeast.

5. The pathway analysis of the short and long genes is not thorough enough. In addition, the authors should use random sets of genes (same number) from the intermediate genes, which are the majority of genes.

We now analyse all the genes in the system, including the intermediate genes. We have divided the human genome into 20 gene sets based on gene length as shown in new Figure 8—figure supplement 2. Our conclusions remain the same, that there are strong patterns of change in proportion of pathways annotated to genes depending on the gene length. We have also included additional plots that show how gene length is distributed among the top most gene-length-dependent pathways, shown in author response image 3, and new Figure 8—figure supplement 3.

6. The time it takes to transcribe a gene is not only dependent on its size and a fixed speed. This is an oversimplification.

We agree it is a simplified assumption, however there are examples where length and the speed of the transcript define the time it takes. For instance, the human Dystrophin gene takes 16hrs to transcribe and this does not appear to vary between measurements (https://pubmed.ncbi.nlm.nih.gov/7719347/). We now include this reference in the paper. We also include a relaxation of this assumption in Figure 3—figure supplement 1, where the transcription rate is allowed to vary (following a normal distribution) and this does not affect our conclusions.

[Editors’ note: what follows is the authors’ response to the third round of review.]

Essential revisions:Please revise your writing to address Reviewer 1 and 3's critiques.Reviewer #1:Authors have mainly addressed my comments.Figure 9: I wonder whether you can make further statements. For example, if immune cells have short cell cycle, then its enrichment for short genes will make more sense.

We tried this, but found it is difficult to make further statements about immune cells for three main reasons. First, they seem to have diverse cell cycle dynamics. For example, CD8^+^ T cells can divide as fast as 2 hours, with most averaging 6 hours with very little G1, though this changes after 5 divisions where the cell cycle slows down (PMID: 21079741). On the other hand, macrophages can have a ~19 h cell cycle (PMID: 3622537). Second, it is challenging to collect comprehensive immune system information to support general statements because most papers with cell cycle duration information don’t usually measure or report it, but instead report related values (such as mitotic rate or doubling time of cell culture) that may or may not be possible to infer cell cycle duration from (e.g. the reported values may be an average over a large population of different cell types). Third, most cell cycle studies examine the regenerating hematopoietic system in adults, rather than immune system development, and these likely have different cell cycle dynamics. In the regenerative case, hematopoietic stem cells are slow dividers and speed up as you go down the lineage (PMID:30084312). For instance in (PMID:30084312), LSK (Lin-Sca-1+c-Kit+) cells, which contain adult bone marrow hematopoietic stem cells (HSCs), have a cell cycle of 47.0 +/-4.9 hrs and give rise to LSK- (Lin-Sca-1+c-Kit-), which have a cell cycle of 23.4+/-2.3 hrs. This also happens for other hematopoietic cell types (PMID:30084312). We are currently working on modeling hematopoiesis to help study the differences between development and regenerative contexts as part of a future project.

Also, might olfactory short genes be related to environmental sensing genes which in turn involve signal transduction pathways also used in fast-growing cells?

The genes contributing to this pattern are almost all olfactory receptors, which are generally very short (e.g. in human, the family of over 500 genes has an average length of around 7000bp, with many in the 1000bp range). These genes are also environmental sensing and signaling genes, though these latter terms are more general and contain many other genes and have a wider gene length distribution, as can be seen in Author response image 4. We clarified this in the manuscript.

**Author response image 4. respfig4:** 

Figure 5 legend: 2^20^ should be 2^19^.

Thank you. We fixed it in the legend.

Reviewer #3:The authors have invested efforts to address the issues that were raised by the reviewers. The story of this manuscript has not fundamentally changed (which probably was also not expected) and there remain shortcomings. One aspect that I wish would improve is to use more understatement rather than claiming things that the authors cannot prove.Here are a few examples, there the manuscript could be improved:1. Line 128/129: "found that, in general, short genes have a higher expression level than longer genes within a cell." When I was reading this, I had trouble believing it but in Figure 3B, the authors show mRNA expression. This is though not mentioned in the text and the reader can be mislead that this also applies to protein expression. It would be desirable that the authors are precise without using generalizations.

We have now specified that we mean transcript (or sometimes more precisely mRNA) expression and transcript counts throughout the manuscript.

2. Line 179: "second child cell" I believe these are usually referred to as "daughter cells".

It is correct that “daughter cells” is the typical term, however, we chose to use gender neutral terminology in our manuscript.

3. Line 233: "We started by asking if organisms with longer genes would also take longer to develop." I apologize but this question (or hypothesis) does not make a lot of sense to me. There are a million reasons why an organism takes a certain amount of time to develop and this may be also dependent on the environment. Reducing it to the length of the genes is surely only one of many reasons. In their conclusion on line 249, the authors call it "strong relationship", which probably is an association and we all know that associations are weak (remember the one about the amount of chocolate consumption and that chance to win the Nobel prize?).

We agree, we have now updated the text to clarify it is an association.

4. Line 266-278: I am not sure if I get the point here "cell cycle duration and gene expression vary spatially.". Not only spatially but also dependent on age, environment, nutrition, and many more factors.

We have clarified that we do not exclude other factors, just that gene length can be one of the mechanisms which helps set up the spatial boundaries within the organism. We analyzed this factor because we had data for it. We have clarified this point in the revision.

5. In the discussion, the limitations (some of which are mentioned) should be discussed much more honestly.

We have tried to reiterate limitations throughout the manuscript and in the discussion.

Here are specific limitations we have now clarified in the discussion:

1) We did not separate the phases of the cell cycle; we only considered the interphase and M phase.

2) We only consider transcription and the theoretical time taken for it and not other aspects of gene/protein expression.

3) We do not consider the effects of gene regulatory networks, cell-cell interactions or other effects that are known to play a role in development.

4) We limited the model to two modes of division: symmetric (where the cell gives rise to identical cells, e.g. Figure 5A) and asymmetric (where the cell gives rise to a fast and slow cell, e.g. Figure 5C).

6. In several figures (for example Figure 6—figure supplement 2 but there are others), the authors use a representation (word clouds) that are not very helpful. The authors should find a better way to bring across the point that they are trying to make.

We tried many ways to visualize the large amount of data we have considered, but each had different advantages and disadvantages and we couldn’t find a single best representation, while maintaining a consistent analysis across all species. For example, representing the data using the typical bar and pie charts has the advantage of being familiar, but these plots were unreadable when all the information was displayed. There is a trade off between the amount of detail that can be shown and the ability to summarize many details in a large data set. We thus decided to include multiple representations in the revision. We kept word clouds to summarize information about gene annotation for thousands of genes per species, and included a pathway enrichment analysis visualization made using the Cytoscape analysis software (Figure 8 supplementary 1), a linear plot with a moving average across bins (Figure 8 supplementary 2) and a matrix plot displaying frequencies of gene annotation themes (Figure 8 supplementary 4) as alternative perspectives and approaches to the same data. Since we wanted to incorporate all Gene Ontology biological processes (pathways) with more than two genes for all species we needed to digest the large data set (13 species, range of 4,700 to 26,000 genes per species, biological pathway annotation range of 2,000 to 12,000 pathways per species, which is millions of data points) and have our message accessible to the readers.